# LEARNING PREDICTIVE CHECKLISTS WITH PROBABILISTIC LOGIC PROGRAMMING

## ABSTRACT

Checklists have been widely recognized as effective tools for completing complex tasks in a systematic manner. Although originally intended for use in procedural tasks, their interpretability and ease of use have led to their adoption for predictive tasks as well, including in clinical settings. However, designing checklists can be challenging, often requiring expert knowledge and manual rule design based on available data. Recent work has attempted to address this issue by using machine learning to automatically generate predictive checklists from data, although these approaches have been limited to Boolean data. We propose a novel method for learning predictive checklists from diverse data modalities, such as images and time series, by combining the power of dedicated deep learning architectures with the interpretability and conciseness of checklists. Our approach relies on probabilistic logic programming, a learning paradigm that enables matching the discrete nature of checklist with continuous-valued data. We propose a regularization technique to tradeoff between the information captured in discrete concepts of continuous data and permit a tunable level of interpretability for the learned checklist concepts. We demonstrate that our method outperforms various explainable machine learning techniques on prediction tasks involving image sequences, medical time series, and clinical notes.

## 1 INTRODUCTION

In recent years, machine learning models have gained popularity in the healthcare domain due to their impressive performance in various medical tasks, including diagnosis from medical images and early prediction of sepsis from clinical time series, among others (Davenport & Kalakota, 2019; Esteva et al., 2019). Despite the proliferation of these models in the literature, their wide adoption in real-world clinical practice remains challenging (Futoma et al., 2020; Ahmad et al., 2018; Ghassemi et al., 2020; De Brouwer et al., 2022). Ensuring the level of robustness required for healthcare applications is difficult for deep learning models due to their inherent black box nature. Non-interpretable models make stress testing arduous and thus undermine the confidence required to deploy them in critical applications such as clinical practice. To address this issue, recent works have focused on developing novel architectures that are both human-interpretable and retain the high performance of black box models (Ahmad et al., 2018).

One such approach is learning medical checklists from available medical records. Due to their simplicity and ability to assist clinicians in complex situations, checklists have become increasingly popular in medical practice (Haynes et al., 2009). However, the simplicity of using checklists typically contrasts with the complexity of their design. Creating a performant checklist requires domain experts who manually collect evidence about the particular clinical problem of interest (Hales et al., 2008), and subsequently reach consensus on meaningful checklist rules (Hales et al., 2008). As the number of available medical records grows, the manual collection of evidence becomes more tedious, bringing the need for partially automated design of medical checklists.

Recent works have taken a step in that direction by learning predictive checklists from Boolean, categorical, or continuous tabular data (Zhang et al., 2021; Makhija et al., 2022). Nevertheless, many available clinical data, such as images or time series, are nor categorical nor tabular by nature. They therefore fall outside the limits of applicability of previous approaches for learning checklists from data. This work aims at addressing this limitations.

Prior work leverages integer programming to generate checklists, but the discrete (combinatorial) nature of solving integer programs makes it challenging to learn predictive checklists from images or time series data. Deep learning architectures rely on gradient-based optimization which differs in style and is difficult to reconcile with integer programming (Shvo et al., 2021). We instead propose to formulate predictive checklists within the framework of probabilistic logic programming. This enables us to extract binary concepts from high-dimensional modalities like images, time series, and text data according to a probabilistic checklist objective, while propagating derivatives throughout the entire neural network architecture. Unlike existing approaches, ProbChecklist doesn't rely on fixed summary extractors such as mean or standard deviation of time series; instead, it learns concepts using neural networks (concept learners).

Our architecture, ProbChecklist, operates by creating binary concepts from high-dimensional inputs, which are then used for evaluating the checklist. However, they are learnt with deep neural networks and are not necessarily interpretable. We therefore investigate two different strategies for providing predictive yet interpretable concepts. The first relies on using inherently interpretable concept extractors, which only focus on specific aspects of the input data (Johnson et al., 2022). The second adds regularization penalties to enforce interpretability in the neural network by design. Several regularization terms have been coined to ensure the concepts are unique, generalizable, and correspond to distinctive input features (Jeffares et al., 2023) (Zhang et al., 2018).

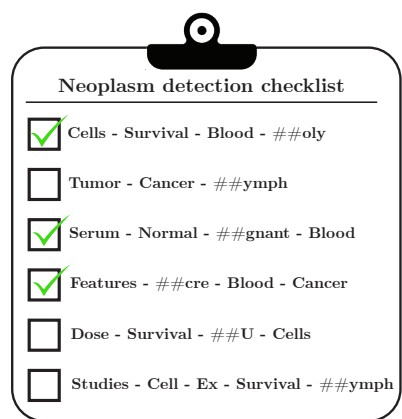

Figure 1: Example checklist learnt by our architecture. Three or more checks entail a positive neoplasm prediction.

Clinical practice is a highly stressful environment where complex decisions with far-reaching consequences have to be made quickly. In this context, the simplicity, robustness, and effectiveness of checklists can make a difference (Hales et al., 2007). Healthcare datasets contain sensitive patient information, including ethnicity and gender, which should not cause substantial differences in the treatment provided. Nevertheless, machine learning models trained on clinical data have been shown to exhibit unacceptable imbalance of performance for different population groups, resulting in biased predictions. When allocating scarce medical resources, fairness should be emphasized more than accuracy to avoid targeting minority subgroups (Fawzy et al., 2022). In an attempt to mitigate this problem, we study the impact of including a fairness regularization into our architecture and report significant reductions in the performance gap across sensitive populations.

We validate our approach empirically on several classification tasks using various data modalities such as images and clinical time series. We show that ProbChecklist outperforms previous learnable predictive checklist approaches as well as several interpretable machine learning baselines. We showcase the capabilities of our method on two healthcare case studies, learning interpretable checklists to early predict the occurrence of sepsis and mortality prediction for intensive care patients.

**Contributions.**

- We propose the first framework to learn predictive checklists from arbitrary input data modalities. Our approach can learn checklists and extract meaningful concepts from time series and images, among others. In contrast with previous works that used (mixed-)integer programming, our approach formulates the predictive checklist learning within the framework of probabilistic logical programming.

- We investigate the impact of different schemes for improving the interpretability of the concepts learnt as the basis of the checklist. We employ regularization techniques to encourage the concepts to be distinct, so they can span the entire input vector and be specialized, i.e. ignore the noise in the signal and learn sparse representations. We also investigate the impact of incorporating fairness constraints into our architecture.

- We validate our learning framework on different data modalities such as images, text and clinical time series, displaying significantly improved performance compared to state-of-the-art checklist learning schemes.

## 2 RELATED WORKS

A major motivation for our work is the ability to learn effective yet interpretable predictive models from data, as exemplified by the interpretable machine learning literature. Conceptually, our method builds upon the recent body of work on learning predictive checklists from data. The implementation of our solution is directly inspired by the literature on probabilistic logic programming.

**Interpretable machine learning.**   Motivated by the lack of robustness and trust of black box models, a significant effort has been dedicated to developing more human-interpretable machine learning models in the last years (Ahmad et al., 2018; Murdoch et al., 2019). Among them, one distinguishes between *intrinsic* (*i.e.* when the model is itself interpretable such as decision trees) and *posthoc* (*i.e.* when trained models are interpreted a posteriori) methods (Du et al., 2019). Checklists belong to the former category as they are an intuitive and easy to use decision support tool. Compared to decision trees, checklists are more concise (there is no branching structure) and can thus be potentially more effective in high stress environments (a more detailed argument is presented in Appendix D). Our approach also relies on building concepts from the input data. Because the concepts are learnt from data, they may themselves lack a clear interpretation. Both intrinsic and posthoc interpretability techniques can then be applied for the concept extraction pipeline (Jeffares et al., 2023). Concept Bottleneck Models (Koh et al., 2020) insert a concept layer before the last fully connected layer, assigning a human-understandable concepts to each neuron. However, a major limitation is that it requires expensive annotated data for predefined concepts.

**Rule-based learning.**   Boolean rule mining and decision rule set learning is a well-studied area that has garnered considerable attention spurred by the demand for interpretable models. Some examples of logic-based models include Disjunctive Normal Forms (OR of ANDs), Conjunctive Normal Forms (AND of ORs), chaining of rules in the form of IF-THEN-ELSE conditions in decision lists, and decision tables. Most approaches perform pre-mining of candidate rules and sample rules using integer programs (IP), simulated annealing, performing local search algorithm for optimizing simplicity and accuracy (Lakkaraju et al., 2016), and Bayesian framework for constructing a maximum a posteriori (MAP) solution (Wang et al., 2017).

**Checklist learning.**   Checklists, pivotal in clinical decision-making, are typically manually designed by expert clinicians (Haynes et al., 2009). Increasing medical records make manual evidence collection tedious, prompting the need for automated medical checklist design. Recent works have taken a step in that direction by learning predictive checklists from Boolean or categorical medical data (Zhang et al., 2021). Makhija et al. (2022) have extended this approach by allowing for continuous tabular data using mixed integer programming. Our work builds upon these recent advances but allows for complex input data modalities. What is more, in contrast to previous works, our method does not rely on integer programming and thus exhibits much faster computing times and is more amenable to the most recent deep learning stochastic optimization schemes.

**Probabilistic logical programming.**   Probabilistic logic reasoning combines logic and probability theory. It represents a refreshing framework from deep learning in the path towards artificial intelligence, focusing on high-level reasoning. Examples of areas relying on these premises include statistical artificial intelligence (Raedt et al., 2016; Koller et al., 2007) and probabilistic logic programming (De Raedt & Kimmig, 2015). More recently, researchers have proposed hybrid architectures, embedding both deep learning and logical reasoning components (Santoro et al., 2017; Rocktäschel & Riedel, 2017; Manhaeve et al., 2018). Probabilistic logic reasoning has been identified as important component for explainable or interpretable machine learning, due to its ability to incorporate knowledge graphs (Arrieta et al., 2020). Combination of deep learning and logic reasoning programming have been implemented in interpretable computer vision tasks, among others (Bennetot et al., 2019; Oldenhof et al., 2023).

## 3 BACKGROUND

**Problem Statement:**  We consider a supervised learning problem where we have access to $N$ input data points $\mathbf{x}_i \in \mathcal{X}$ and corresponding binary labels $y_i \in \{0, 1\}$. Each input data point consists of a collection of $K$ data modalities: $\mathbf{x}_i = \{\mathbf{x}_i^1, \mathbf{x}_i^2, \dots, \mathbf{x}_i^K\}$. Each data modality can either be continuous ($\mathbf{x}_i^k \in \mathbb{R}^{d_k}$) or binary ($\mathbf{x}_i^k \in \{0, 1\}^{d_k}$). Categorical data are assumed to be represented in expanded binary format. We set $d$ as the overall dimension of $\mathbf{x}_i$. That is, $d = \sum_{k=1}^{K} d_k$. The $N$ input data points and labels are aggregated in a data structure $\mathbf{X}$ and a vector $\mathbf{y}$ respectively.

Our objective is to learn an interpretable decision function $f : \mathcal{X} \to \{0,1\}$ from some domain $\mathbb{F}$ that minimizes some error criterion $d$ between the predicted and the true label. The optimal function $f^*$ then is: $f^* = \arg\min_{f \in \mathcal{F}} \mathbb{E}_{\mathbf{x},\mathbf{y} \sim \mathcal{D}}[d(f(\mathbf{x}), \mathbf{y})]$, where $\mathcal{D}$ stands for the observational data distribution. We limit the search space of decision functions $\mathcal{F}$ to the set of predictive checklists, which are defined below.

**Predictive checklists:** Generally, we define a predictive checklist as a linear classifier applying on a list of $M$ binary concepts $\mathbf{c}_i \in \{0,1\}^M$. A checklist will predict a data point, consisting of $M$ concepts $\mathbf{c}_i = \{c_i^1, \ldots, c_i^M\}$, as positive if the number of concepts such that $c_i^m = 1$ is larger or equal to a threshold $T$. That is, given a data point with concepts $\mathbf{c}_i$, the predicted label of a checklist with threshold $T$ is expressed as: $\hat{y}_i = \begin{cases} 1 & \text{if} \quad \sum_{m=1}^{M} c_i^m \geq T \\ 0 & \text{otherwise} \end{cases}$

The only parameter of a checklist is the threshold $T$. Nevertheless, the complexity lies in the definition of the list of concepts that will be given as input to the checklist. This step can be defined as mapping $\phi$ that produces the binary concepts from the input data: $\mathbf{c}_i = \psi(\mathbf{x}_i)$. Existing approaches for learning checklists from data differ by their mapping $\psi$. Zhang et al. (2021) assume that the input data is already binary. In this case, the mapping $\psi_M$ is then a binary matrix $\Psi \in \{0,1\}^{M \times k}$ such that $\Psi \mathbf{1}_k = \mathbf{1}_k$, where $\mathbf{1}_k$ is a column vector of ones[1]. One then computes $\mathbf{c}_i$ as $\mathbf{c}_i = \Psi_M \mathbf{x}_i$. The element of $\Psi_M$ as well as the number of concepts $M$ (hence the dimension of the matrix) are learnable parameters.

Previous approaches (Makhija et al., 2022) relax the binary input data assumption by allowing for the creation of binary concepts from continuous data through thresholding. Writing $\mathbf{x}_i^b$ and $\mathbf{x}_i^c$ for the binary and real parts of the input data respectively, the concept creation mechanism transforms the real data to binary with thresholding and then uses the same matrix $\Psi_M$. We have $\mathbf{c}_i = \Psi_M[\mathbf{x}_i^b, \text{sign}(\mathbf{x}_i^b - \mathbf{t}_i)]$, where $[\cdot, \cdot]$ is the concatenation operator, $\mathbf{t}_i$ is a vector of thresholds, $\text{sign}(\cdot)$ is an element-wise function that returns $1$ is the element is positive and $0$ otherwise. In this formulation one learns the number of concepts $M$, the binary matrix $\Phi_M$ as well as the thresholds values $\mathbf{t}_i$.

**Probabilistic Logic Programming:**

Probabilistic logical reasoning is a knowledge representation approach that involves the use of probabilities to encode uncertainty in knowledge. This is encoded in a probabilistic logical program (PLP) $\mathcal{P}$ connected by a set of $N$ probabilistic facts $U = \{U_1, ..., U_N\}$ and $M$ logical rules $F = \{f_1, ...f_M\}$. PLP enables inference on knowledge graphs $\mathcal{P}$ by calculating the probability of a query. This query is executed by summing over the probabilities of different "worlds" $w = u_1, ..., u_N$ (i.e., individual realizations of the set of probabilistic facts) that are compatible with the query $q$. The probability of a query $q$ in a program $\mathcal{P}$ can be inferred as $P_{\mathcal{P}}(q) = \sum_w P(w) \cdot \mathbb{I}[F(w) \equiv q]$, where $F(w) \equiv q$ indicates that the propagation of the realization $w$ across the knowledge graph, according to the logical rules $F$, leads to $q$ being true. The motivation behind using PLP is to navigate the tradeoff between discrete checklists and learnable soft concepts. Incorporating a neural network into this framework enables the generation of probabilistic facts denoted as the neural predicate $U^\theta$, where $\theta$ represents the weights. These weights can be trained to minimize a loss that depends on the probability of a query $q$: $\hat{\theta} = \arg\min_\theta \mathcal{L}(P(q \mid \theta))$.

## 4 PROBCHECKLIST: LEARNING FAIR AND INTERPRETABLE PREDICTIVE CHECKLISTS

### 4.1 ARCHITECTURE OVERVIEW

Our method first applies concept extractors on each data modality. Each concept extractor outputs a list of concept probabilities for each data modality. These probabilities are then concatenated to form a vector of probabilistic concepts ($\mathbf{p_i}$) for a given data sample. This vector is dispatched to a probabilistic logic module that implements a probabilistic checklist with query $q := \mathcal{P}(\mathbf{y_i} = \hat{\mathbf{y}_i})$. We can then compute the probability of the label of each data sample and backpropagate through the whole architecture. At inference time, the checklist inference engines discretize the probabilistic

---

[1]This corresponds effectively to every row of $\Psi$ summing to 1.

checklist to provide a complete predictive checklist. A graphical depiction of the overall architecture is given in Figure 2.

## 4.2 DATA MODALITIES AND CONCEPTS

Data modalities refer to the distinct sets of data that characterize specific facets of a given process. For instance, in the context of healthcare, a patient profile typically includes different clinical time series, FMRI and CT-scan images, as well as prescriptions and treatment details in text format. The division in data modalities is not rigid but reflects some underlying expert knowledge. Concepts are characteristic binary variables that are learnt separately for each modality.

## 4.3 CONCEPT EXTRACTOR

Instead of directly learning binary concepts, we extract soft concepts that we subsequently discretize. For each of the $K$ data modalities, we have a soft concept extractor $\psi_k : \mathbb{R}^{d_k} \to [0,1]^{d'_k}$ that maps the input data to a vector of probabilities $\mathbf{p}_i^k$, where $d'_k$ is the number of soft concepts to be

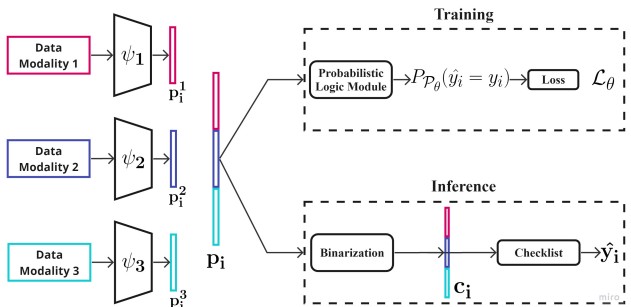

Figure 2: **Overview of our proposed ProbChecklist.** Given $K$ data modalities as the input for sample $i$, we train $K$ concept learners to obtain the vector of probabilistic concepts of each modality $\mathbf{p}_i^k \in [0,1]^{d'_k}$. Next, we concatenate into the full concepts probabilities ($\mathbf{p}_i$) for sample i. For training the concept learners, we pass $\mathbf{p}_i$ through the probabilistic logic module. At inference time, we discretize $\mathbf{p}_i$ to construct a complete predictive checklist.

extracted from data modality $k$. Concatenating the outputs of the $K$ concept extractors results in a vector of probabilities $\mathbf{p}_i \in [0,1]^{d'}$, with the $d'$ the total number of soft concepts.

## 4.4 CHECKLIST LEARNING

The checklist prediction formula of Equation 3 can be understood as logical rules in a probabilistic logical program. Together with the probabilities of each concepts, encoded in vector $\mathbf{p}_i$ that represent $d'$ probabilistic facts, this represents a probabilistic logical program $\mathcal{P}_\theta$. We refer to $\theta$ as the set of learnable parameters in the probabilistic logical program.

We want to maximize the probability of a the prediction being correct. That is, we want to maximize the probability of the query $q := \hat{y}_i = y_i$,

$$\hat{\theta} = \arg\min_\theta -P_{\mathcal{P}_\theta}(\hat{y}_i = y_i) = \arg\min_\theta -\sum_w P(w) \cdot \mathbb{I}[F(w) \equiv (\hat{y}_i = y_i)] \tag{1}$$

By interpreting the probabilities $\mathbf{p}_i$ as the probability that the corresponding binary concepts are equal to 1 (*i.e.* $\mathbf{p}_i[j] = P(\mathbf{c}_i[j] = 1)$, where $[j]$ indexes the $j$-th component of the vector), we can write the probability of query $q$ as follows.

**Proposition 4.1.** *The probability of the query $\hat{y}_i = y_i$ in the predictive checklist is given by*

$$P_{\mathcal{P}_\theta}(\hat{y}_i = 1) = 1 - P_{\mathcal{P}_\theta}(\hat{y}_i = 0) = \sum_{d=T}^{d'} \sum_{\sigma \in \Sigma^d} \prod_{j=1}^{d'} (\mathbf{p}_i[j])^{\sigma(j)} (1 - \mathbf{p}_i[j])^{1-\sigma(j)} \tag{2}$$

*where $\Sigma_d$ is the set of selection functions $\sigma : [d'] \to \{0, 1\}$ such that $\sum_{j=1}^{d'} \sigma(j) = d$.*

The detailed derivations are presented in Appendix A. We use the log-likelihood as the loss function, which leads our final loss: $\mathcal{L} = y_i \log(P_{\mathcal{P}_\theta}(\hat{y}_i = 1)) + (1 - y_i) \log(P_{\mathcal{P}_\theta}(\hat{y}_i = 0))$.

The parameters $\theta$, include multiple elements: the parameters of the different soft concept extractors ($\theta_\psi$), the number of concepts to be extracted for each data modality $d'_k$, the checklist threshold $T$. As the soft concept extractors are typically parameterized by neural networks, optimizing $\mathcal{L}$ with respect to $\theta_\psi$ can be achieved via gradient based methods. $d'_k$ and $T$ are constrained to be integers and are thus treated as hyper-parameters in our experiments.

## 4.5 CHECKLIST INFERENCE

ProbChecklist relies on soft concepts extraction for each data modality. Yet, at test time, a checklist operates on binary input data. We thus binarize the predicted soft concepts by setting $\mathbf{c}_i[j] =$

$\mathbb{I}[\mathbf{p}_i[j] > \tau]$. The thresholding parameter $\tau$ is an hyperparameter that can be tuned based on validation data. After training, we construct the final checklist by pruning the concepts that are never used in the training data (*i.e.* concepts $j$ such that $\mathbf{c}_i[j] = 0, \forall i$, are pruned). This step offers users the flexibility to tune between sensitivity-specificity depending on the application. The optimal checklist can be obtained by varying $\tau$ to optimize the desired metric on the validation data.

### 4.6 INTERPRETABILITY OF THE CONCEPT EXTRACTORS

The checklist concepts are learnt using deep neural networks and are, therefore, not interpretable in general. To address this issue, we propose two mechanisms to improve the interpretability of the learnt concepts: *focused* concept learners and regularization terms that incorporate explainability in the structure of the concept learners. Focused models limit the range of features that can contribute to a concept. This can be done via manual specification of the models e.g. using different LSTMs for each time series (Johnson et al., 2022). Regularization terms such as TANGOS helps unveil each input signal's contribution to the learnt concepts for a given sample, making the concepts interpretable. It ensures that the concepts are obtained from distinct and sparse subsets of the input vector, avoiding overlap. Sparsity is achieved by taking the L1-norm of the concept gradient attributions with respect to the input vector. To promote decorrelation of signal learned in each concept, the loss is augmented by incorporating the inner product of the gradient attributions for all pairs of concepts. This scheme compels the models to learn unique concepts. More details about TANGOS and its mathematical formulation can be found in the Appendix F.1. We additionally introduce a regularization term which propels the learnt concept probabilities towards either 0 or 1. This term helps in identifying characteristic concepts for each patient: $\mathcal{L}_{prob-reg} = \sum_j \sum_i p_i[j]$.

### 4.7 FAIRNESS REGULARIZATION

We encourage fairness of the learnt checklists by equalizing the error rate across subgroups of protected variables. This is achieved by penalizing significant differences in False Positive and False Negative Rates for sensitive subgroups (Pessach & Shmueli, 2022). For a binary classification problem, with protected attribute $S$, predicted labels $\hat{y} \in \{0, 1\}$, and actual label $y \in \{0, 1\}$, we define separations as follows (Corbett-Davies & Goel, 2018):

$$\Delta FPR = \|P(\hat{y}_i = 1 | y = 0, S = s_i) - P(\hat{y}_i = 1 | y = 0, S = s_j)\|_1 \quad \forall s_i, s_j \in S \qquad (3)$$

$$\Delta FNR = \|P(\hat{y}_i = 0 | y = 1, S = s_i) - P(\hat{y}_i = 0 | y = 1, S = s_j)\|_1 \quad \forall s_i, s_j \in S \qquad (4)$$

and combine these in a fairness regularizer $\mathcal{L}_{\text{Fair}} = \lambda(\Delta FPR + \Delta FNR)$.

## 5 EXPERIMENTS

We investigate the performance of ProbChecklist along multiple axes. We first compare the classification performance against a range of interpretable machine learning baselines. Second, we investigate the importance of several key hyperparameters of our method. Lastly, we demonstrate how we can tune the interpretability of the learnt concepts and how we can enforce fairness constraints into ProbChecklist. Complete details about the datasets, baselines used in our experiments, and hyperparameter tuning are available in Appendix E, C.

**Baselines.** We compare our method against the following baselines.

*Mixed Integer Programming (MIP)*(Makhija et al., 2022). This approach allows to learn predictive checklists from continuous inputs. For images or time series, we typically apply MIP on top of an embedding obtained from a pre-trained deep learning model.

*Integer Linear Program (ILP)*(Zhang et al., 2021). ILP learns predictive checklists with Boolean inputs. We apply these to tabular data by categorizing the data using feature means as threshold.

*CNN/LSTM/BERT + Logistic Regression (LR)*. This consists in using a CNN, LSTM or pre-trained BERT on the input data and applying a logistic regression on the combination of the last layer's embeddings of each modality.

*CNN/LSTM/BERT + Multilayer perceptron (MLP)*. This is similar to the previous approach but where we apply an MLP on the combination of the last layer's embeddings of each modality.

**Datasets.** A crucial strength of our method resides in its ability to learn predictive from high dimensional input data. We briefly describe the MNIST synthetic dataset created here and defer the descriptions of other datasets (PhysioNet sepsis tabular dataset, MIMIC mortality dataset, Medical Abstracts TC Corpus) to the Appendix E.3

*Synthetic MNIST checklist.* Due to the absence of real-world datasets with ground-truth checklists, we first validate our idea on a synthetic setup created using MNIST image sequences as input and a checklist defined on digit labels. Each sample consists of a sequence of $\mathbf{K} = 4$ MNIST images (treating each image as a separate modality). We then assign a label to *each samples* according to the following ground-truth checklist. (i) Digit of $\mathbf{Image\,1} \in \{0, 2, 4, 6, 8\}$, (ii) $\mathbf{Image\,2} \in \{1, 3, 5, 7, 9\}$, (iii) $\mathbf{Image\,3} \in \{4, 5, 6\}$, (iv) $\mathbf{Image\,4} \in \{6, 7, 8, 9\}$. If at least 3 of the rules are satisfied, the label is 1, and 0 otherwise.

## 5.1 CHECKLIST PERFORMANCE

We evaluate the classification performance of the different models according to accuracy, precision, recall and specificity. For the checklist baselines, we also report the total number of concepts used ($M$) and the threshold for calling a positive sample ($T$). Results are presented in table 1. Additional results and details about hyperparameter tuning are provided in the Appendix E,C.

| Dataset | Model | Accuracy | Precision | Recall | Specificity | $d'_k$ | T | M |
|---|---|---|---|---|---|---|---|---|
| MNIST Checklist | CNN + MLP | 94.72 ± 4.32 | 0.895 ± 0.1 | 0.835 ± 0.13 | 0.976 ± 0.02 | | - | - |
| | CNN + LR | 95.04 ± 0.31 | 0.914 ± 0.01 | 0.836 ± 0.016 | **0.98 ± 0.003** | 4 | - | - |
| | pretrained CNN + MIP | 79.56 | 0 | 0 | 1 | | 8 | 13.5 ± 0.5 |
| | **ProbChecklist** | **96.808 ± 0.24** | **0.917 ± 0.015** | **0.929 ± 0.01** | 0.978 ± 0.004 | 4 | 8.4 ± 1.2 | 16 |
| PhysioNet Tabular | Logistic Regression | 62.555 ± 1.648 | 0.624 ± 0.0461 | 0.144 ± 0.0393 | **0.9395 ± 0.0283** | | - | - |
| | Unit Weighting | 58.278 ± 3.580 | 0.521 ± 0.093 | 0.4386 ± 0.297 | 0.6861 ± 0.251 | 1 | 3.2 ± 1.16 | 9.6 ± 0.8 |
| | ILP mean thresholds | 62.992 ± 0.82 | 0.544 ± 0.087 | 0.1196 ± 0.096 | 0.9326 ± 0.0623 | | 2.8 ± 0.748 | 4.4 ± 1.01 |
| | MIP Checklist | **63.688 ± 2.437** | 0.563 ± 0.050 | **0.403 ± 0.082** | 0.7918 ± 0.06 | | 3.6 ± 0.8 | 8 ± 1.095 |
| | **ProbChecklist** | 62.579 ± 2.58 | **0.61 ± 0.076** | 0.345 ± 0.316 | 0.815 ± 0.185S | 1 | 3.6 ± 1.2 | 10 |
| MIMIC III | Unit Weighting | 73.681 ± 0.972 | 0.469 ± 0.091 | 0.223 ± 0.206 | 0.889 ± 0.026 | | 6.1 ± 0.830 | 8.9 ± 0.627 |
| | ILP mean thresholds | 75.492 ± 0.318 | 0.545 ± 0.028 | 0.142 ± 0.059 | 0.959 ± 0.019 | | 3.6 ± 0.894 | 3.6 ± 0.894 |
| | MIP Checklist | 74.988 ± 0.025 | 0.232 ± 0.288 | 0.014 ± 0.017 | **0.997 ± 0.004** | 1 | 4.5 ± 2.082 | 4.5 ± 2.082 |
| | LSTM + LR | 66.585 ± 2.19 | 0.403 ± 0.02 | **0.684 ± 0.039** | 0.66 ± 0.034 | | - | - |
| | LSTM + MLP | 76.128 ± 0.737 | 0.446 ± 0.223 | 0.23 ± 0.132 | 0.939 ± 0.036 | | - | - |
| | LSTM + MLP (all features) | **80.04 ± 0.598** | 0.328 ± 0.266 | 0.129 ± 0.131 | 0.962 ± 0.043 | | - | - |
| | **ProbChecklist** | 77.58 ± 0.481 | **0.642 ± 0.075** | 0.247 ± 0.032 | 0.953 ± 0.019 | 2 | 9.6 | 20 |
| Medical Abstracts Corpus | BERT + ILP | 72.991 ± 8.06 | 0.292 ± 0.29 | 0.197 ± 0.26 | 0.879 ± 0.17 | | 1.2 ± 0.4 | 1.2 ± 0.4 |
| | BERT + MIP | 69.32 ± 8.1 | 0.583 ± 0.14 | 0.059 ± 0.08 | 0.991 ± 0.09 | 6 | 2.5 ± 0.6 | 4 ± 0.8 |
| | BERT + LR | 80.193 ± 0.88 | **0.790 ± 0.051** | 0.138 ± 0.065 | **0.988 ± 0.007** | | - | - |
| | BERT + MLP | 81.782 ± 0.31 | 0.941 ± 0.04 | 0.07 ± 0.009 | 0.961 ± 0.01 | | - | - |
| | **ProbChecklist** | **83.213 ± 0.23** | 0.616 ± 0.006 | **0.623 ± 0.01** | 0.891 ± 0.003 | 6 | 3 | 6 |

Table 1: Performance results for all the models and baselines on all the datasets. We report accuracy, precision, recall as well as conciseness of the learnt checklist. To facilitate visualization and comparison, we plot these results in Section I of the Appendix (Figure 10).

**MNIST Checklist.** We used a simple three-layered CNN model as the concept learner for each image. In Table 1, we report the results of the baselines and ProbChecklist for $\mathbf{d'_k = 4}$ ($\mathbf{M = 16}$) on the test samples. Our method outperforms all the baselines, in terms of accuracy and recall, indicating that it identifies the minority class better than these standard approaches. The MIP failed to find solutions for some folds of the dataset and didn't generalise well on the test samples.

**Sepsis Prediction from Tabular Data.** This setup is ideal for comparison with existing checklist method as they only operate on tabular dataset. In Figure 4, we visualize learnt by ProbChecklist in one of the experiments. We observe that ProbChecklist exhibits similar performance to checklist baselines. We want to emphasize that ProbChecklist provides a significantly broader applicability to multimodal datasets while maintaining comparable performance on tabular datasets, thus making it valuable.

**Neoplasm Detection from Clinical Abstracts.** We use a pretrained BERT model (Alsentzer et al., 2019) with frozen weights as our concept learner. This was a BioBERT model pretrained on clinical notes of MIMIC-III dataset. Our checklist has a much better recall and accuracy than previous methods. Both checklist learning and deep learning methods give poor performance on the minority class.

**Mortality Prediction using Time Series Data.** To learn representations from clinical timeseries, we initialize $K$ two-layered LSTMs. We highlight our key results in Table 1. For ProbChecklist, we report the checklist which attains the highest accuracy on validation data. We surpass existing methods in terms of accuracy and precision with a significant margin. We find that a checklist with better recall can be constructed by optimizing over F1-Score instead of accuracy.

**Sensitivity analysis:** We investigate the evolution of performance of ProbChecklist with increasing number of learnt concepts $\mathbf{d'_k}$. On Figure 3a, we show the accuracy, precision, recall, and specificity

in function of the number of concepts per image on the MNIST dataset. We observe a significant improvement in performance when $d'_k$ increases from $1$ to $2$, which suggests that having learning one concept per image is inadequate to capture all the signal in the sample. It is also interesting to note that the performance reaches a saturation point after $d'_k = 3$. This suggests held-out loss can be used to tune the value of $d'_k$ to find the optimal number of concepts for a given data modality.

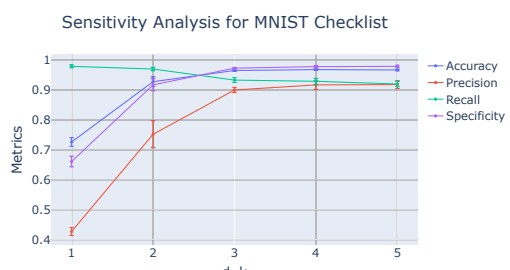

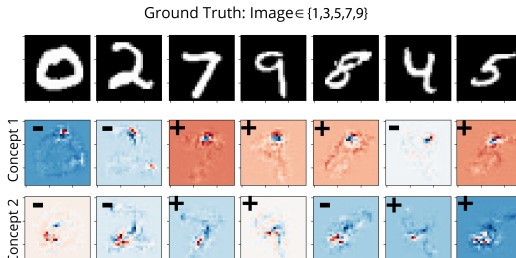

(a) Performance of ProbChecklist with varying $d'_k$ on MNIST Checklist Dataset

(b) We plot images and corresponding gradient attributions heat maps for seven inputs samples of the Image 2 modality of the MNIST dataset. We used a checklist with two learnable concepts per image. The intensity of red denotes the positive contribution of each pixel, whereas blue indicates the negative. If a concept predicted as true for an image, then we represent that with plus (+) sign, and with a negative sign (-) otherwise.

Figure 3

## 5.2 CONCEPTS INTERPRETATION

We investigate the concepts learnt from image and timeseries datasets with an interpretability regularization as in Section 4.6. To gain insight into what patterns of signals refer to each individual concept, we examine the gradient of each concept with respect to each dimension of the input signal. Intuitively, the interpretability regularization enforces the concepts to focus on a sparse set of features of the input data.

**MNIST Images:** We analyze the gradient of our checklist on individual pixels of the input images. We use a checklist with two concepts per image. On Figure 3b, we show example images of the *Image 2* of the MNIST dataset along with the gradient heat map for each learnt concept of the checklist. The ground truth concept for this image is **Image 2** $\in \{1, 3, 5, 7, 9\}$. First, we see that the $7, 9$, and $5$ digits are indeed the only ones for which the predicted concepts of our checklist are positive. Second, we infer from the gradient heat maps that concepts 1 and 2 focus on the image's upper half and centre region, respectively. Concept 1 is true for digits 5, 8, 9 and 7, indicating that it corresponds to a horizontal line or slight curvature in the upper half. Since Digits 0 and 2 have deeper curvature than the other images, and there is no activity in that region in the case of 4, concept 1 is false for them. concept 2 is true for images with a vertical line, including digits 9, 4, 5 and 7. Therefore, concept 2 is false for the remaining digits (0, 2, 8). The checklist outcome matches the ground truth when both concepts are true for a given image. Complementary analyses on MNIST and MIMIC III are provided in Appendix F.2 and F.3. This analysis ensures interpretability at the individual sample level. As illustrated in the previous example, recognizing and comprehending these concepts at the dataset level relies on visual inspection.

**Medical Abstracts:** Compared to images and time series, interpreting concepts learned from textual data is easier because its building blocks are tokens which are already human understandable. For the Neoplasm detection task, we adopt an alternative method by conducting a token frequency analysis across the entire dataset. This approach has yielded a more lucid checklist shown in Figure 1. We identified key tokens associated with positive and negative concepts (positive and negative tokens). Each concept is defined by the presence of positive words and the absence of negative words.

## 5.3 FAIRNESS

We evaluate the fairness of ProbChecklist on the MIMIC-III Mortality Prediction task and show that we can reduce the performance disparities between sensitive attributes by incorporating fairness regularization (FR) terms, as introduced in Section 4.7. We set the sensitive features as gender

$\in \{Male, Female\}$ and ethnicity $\in \{Black, White, Others\}$. Our results are displayed on Tables 2 and 11. These disparities in performance across different sub-populations are significantly reduced after fairness regularization is used. To see the effectiveness of the regularizer, we report the percentage decrease in $\Delta$FNR and $\Delta$FPR observed with respect to the unregularized checklist predictions for all pairs of sensitive subgroups. Similar fairness constraints (FC) can also be added to the ILP mean-thresholds baseline (Jin et al., 2022). We include a separate constraint for each pair that restricts |$\Delta$FNR| and |$\Delta$FPR| to be less than $\epsilon = 0.05$.

It is important to note that our approach minimizes the summation of $\Delta$FNR and $\Delta$FPR across all pairs of subgroups, but in the ILP we can specify a strict upper bound for each pair. Due to this, we might observe an increase in the gap for certain pairs in case of ProbChecklist, but adjusting the relative weights of these terms in the loss equation helps in achieving optimal performance. Although ProbChecklist had higher initial FNR/FPR values, the regularizer effectively reduces them to be comparable to those of ILP, particularly for the ethnicity pairs.

**Sepsis if 3+ items are true**

☐ **Bilirubin direct sd** $\geq 1.31$

☐ **FiO2 sd** $\geq 0.029$

☐ **FiO2 mean** $\geq 0.035$

☐ **EtCO2 sd** $\geq 0$

☐ **FiO2 last** $\geq 0.037$

☐ **Alkalinephos sd** $\geq 0$

☐ **AST sd** $\geq 0$

☐ **pH sd** $\geq 0.221$

☐ **TroponinI sd** $\geq 0$

☐ **Bilirubin direct last** $\geq 7.352$

Figure 4: Learnt checklist for PhysioNet Sepsis Prediction Task (Tabular). We report the performance result as accuracy (65.69%), precision (0.527), recall (0.755), and specificity (0.6).

# 6 DISCUSSION

**Performance of ProbChecklist.** Through these experiments, we aim to showcase that ProbChecklist surpasses existing checklist methods and achieves comparable performance to MLP (non-interpretable) methods. The switch to learnable concepts explains the improvement in accuracy over checklist methods. These concepts capture more signal than fixed summary/concept extractors used in prior works to create binarized tabular data. It's important to note that a checklist, due to its binary weights, has a strictly lower capacity and is less expressive than deep learning but possesses a more practical and interpretable structure. Despite this, it exhibits similar performance to an MLP.

**Interpretability of checklist structure and learnt concepts.** Although ProbChecklist employs a probabilistic objective for training the concept learners, the end classifier used for inference is, in fact, a discrete checklist. While this makes the classifier highly interpretable, it also shifts the focus of interpretability to the learnt concepts. We fully realize this trade-off and investigate existing techniques to maintain feature-space interpretability. For time series and images, we employ regularization terms (4.6) to enforce sparsity, avoid redundancy, and learn strongly discriminative features with high probability. We also use focused concept learners to avoid learning concepts that are functions of multiple modalities. Identifying patterns from the binarized concepts is primarily based on visual inspection and expert knowledge. We noticed it is easier to source and comprehend the key tokens contributing to each concept for text data. Lastly, we want to highlight that ProbChecklist is a flexible framework, and other interpretable models can be easily integrated as concept learners.

| | Method | Female-Male | | White-Black | | Black-Others | | White-Others | |
|---|---|---|---|---|---|---|---|---|---|
| | | $\Delta$FNR | $\Delta$FPR | $\Delta$FNR | $\Delta$FPR | $\Delta$FNR | $\Delta$FPR | $\Delta$FNR | $\Delta$FPR |
| | w/o FC | 0.038 | 0.011 | 0.029 | 0.026 | 0.152 | 0.018 | 0.182 | 0.045 |
| ILP mean thresholds | FC | 0.011 | 0.001 | 0.031 | 0.008 | 0.049 | 0.016 | 0.017 | 0.0007 |
| | %↓ | 71.053 | 90.909 | -6.897 | 69.231 | 67.763 | 11.111 | 90.659 | 98.444 |
| | w/o FR | 0.127 | 0.311 | 0.04 | 0.22 | 0.02 | 0.273 | 0.02 | 0.053 |
| ProbChecklist | FR | 0.103 | 0.089 | 0.028 | 0.016 | 0.021 | 0.008 | 0.006 | 0.008 |
| | %↓ | 18.898 | 71.383 | 30.000 | 92.727 | -5.000 | 97.033 | 70.000 | 85.660 |

Table 2: Improvement in fairness metrics across gender and ethnicity on MIMIC III for the mortality prediction task after adding fairness regularization. We report $\Delta$FNR and $\Delta$FPR for all pairs of subgroups of sensitive features and the percentage decrease (% ↓) wrt unregularized checklist.

**Limitations.** We have taken the first step towards learning checklists from complex modalities, whereas the existing methods are restricted to tabular data. Even though we have a mechanism to learn interpretable checklist classifiers using logical reasoning, more work is needed on the interpretability of the learnt concepts. Another drawback is the exponential memory complexity of the training. A fruitful future direction would be to study approximations to explore a smaller set of combinations of concepts. Detailed complexity analysis can be found in Appendix B.

**Societal Impact.** As discussed initially in the paper, manually designed checklists are extensively used in hospitals for decision-making under complex situations and help automate certain aspects of the treatment. With more research on the interpretability of concepts, ProbChecklist can replace the existing manual procedure and reduce the burden on the healthcare system.

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
