# Appendix

## A  DERIVATIONS FOR THE LOSS FUNCTION

Given the individual probabilities of each concept $j$ being positive, $\mathbf{p}_i[j]$, the probability that the checklist outputs exactly $d$ positive concepts for sample $i$ is given by the binomial distribution:

$$P_{\mathcal{P}_\theta}(\#\text{ true concepts} = d) = \sum_{\sigma \in \Sigma^d} \prod_{j=1}^{d'} (\mathbf{p}_i[j])^{\sigma(j)} (1 - \mathbf{p}_i[j])^{1-\sigma(j)},$$

where $\Sigma_d$ is the set of selection functions $\sigma : [d'] \to \{0, 1\}$ such that $\sum_{j=1}^{d'} \sigma(j) = d$. The probability that the checklist gives a positive prediction is the probability that the number of positive concepts is larger than $T$, we thus have

$$P_{\mathcal{P}_\theta}(\hat{y}_i = 1) = \sum_{d=T}^{d'} \sum_{\sigma \in \Sigma^d} \prod_{j=1}^{d'} (\mathbf{p}_i[j])^{\sigma(j)} (1 - \mathbf{p}_i[j])^{1-\sigma(j)}.$$

## B  COMPLEXITY ANALYSIS

Let $d$ be the number of positive concepts for sample $i$. Probabilistic Checklist Objective is defined as $P(d \geq T)$, i.e. the sample is classified as positive if $T$ or more concepts (out of $M$) are true. We discuss the space and time complexity for training and inference separately.

**Step-wise Breakdown for Training.**

- **Learning concepts probabilities using a concept learner.** The computational complexity is contingent upon the specific neural network employed. In our analysis, we utilize LSTMs, CNNs, and pre-trained BERT models with fixed weights, ensuring that computations can be performed in polynomial time.

- **Loss computation.** We need to compute the probabilistic query $\mathbf{P}(\mathbf{d} \geq \mathbf{T})$. This involves iterating over all possible $2^M$ combinations of true concepts $\forall\, d \geq T$. The probability of each combination is obtained by taking product of concept probabilities.
  Worst case time complexity: $\mathbf{O(M2^M)} \implies$ **exponential**.

We implement the time-efficient $\mathbf{O(1)}$ version of this method by caching the combination tensor in memory. In this situation, the memory cost becomes $\mathbf{O(M2^M)}$. The dimensions of tensor that stores all the combinations is $[2^M, M]$. This is an acceptable trade-off as it enables us to run experiments in reasonable time.

**Inference.** Inference involves a single pass through the concept learner. Since loss does not need to be computed, the inference time is the same for both memory- and time-efficient implementations of ProbChecklist and has the same complexity as the forward propagation of the concept learner.

**Training Details.**

We trained the time-efficient implementation with memory complexity $O(M * 2^M)$ on one NVIDIA A40 GPU with 48GB memory which can fit up to $M = 30$. Exponential complexity is primary reason for performing feature selection and limiting the modalities to 10 and learning up to 3 features per modality.

| Training Time (seconds) | Number of concepts | MIP Checklist | ProbChecklist |
|---|---|---|---|
| PhysioNet Tabular ($d'_k = 1$) | 10 | $667 \pm 80.087$ | $995.0 \pm 35.014$ |
| MNIST ($d'_k = 4$) | 16 | 3600 | $2773 \pm 22.91$ |

Table 3: ProbChecklist exhibits exponential complexity. To gauge the impact of this limitation, we provide training times (in seconds) for both MIP Checklist and ProbChecklist. It's crucial to highlight that while MIP Checklist performs effectively with tabular data, successfully uncovering the optimal solution, its performance is poor when applied to MNIST synthetic setup. Even when we set the runtime for Gurobi solver as 1 hour, it struggles to achieve optimal solutions for many cases. On the other hand, ProbChecklist stands out as more reliable, capable of performing end-to-end training and successfully learning the optimal solution.

A fruitful future direction would be to study approximations to explore a smaller set of combinations.

## C  HYPERPARAMETER TUNING

ProbChecklist framework allows experts to design user-centric checklists. Hyperparameters $d'_k$, $T$ and $M$ represent the structure/compactness of the checklist but alone aren't sufficient to garner information about the checklist's performance. Different $d'_k$ are tried for each modality in an increasing fashion to find the one that performs best. Sensitivity analysis to study the relation between

$d'_k$ and performance (Figure 3a) suggests that the performance saturates after a certain point. This point can be determined experimentally for different modalities. M, total concepts in the checklist, is obtained by pruning concepts that are true for insignificant samples. Our experiments showed that pruning was not required since all the concepts were true for a significant fraction of samples. This indicates the superior quality of the concepts.

We try different values of T in the range [M/4, M/2] (total items M) to find the most performant model. However, this hyperparameter tuning doesn;t contribute to the computational cost. For $d'_k$, we only only 2-3 values, and use the same value for all the features. Domain experts will be more equipped to choose these values based on their knowledge of the features. For example, if we are recording a time series feature known to stay stable (not fluctuate much), then low $d'_k$ is sufficient. The value of $d'_k$ also depends on the number of observations (most blood tests aren't performed hourly, but heart rate and oxygen saturation are monitored continuously).

# D    Conciseness of checklists over decision trees

Decision trees represent another highly interpretable classification method. Compared to checklists, they strike a different balance between expressivity and conciseness. While trees grow exponentially with the number of rules, checklists only grow linearly. Yet, the increased complexity of trees makes them also more expressive. Each decision in a tree only applies to its specific lineage. In contrast, each rule in a checklist applies globally. Despite their lower expressivity, the conciseness of checklists can make them more practical and interpretable than decision trees.

# E    Experiment Details

In this section we provide additional details about the experiments.

## E.1    Baselines

We compare the performance of our method against standard classifiers, including logistic regression (LR) and a classical multilayer perceptron (MLP). For a fair comparison, the process to obtain the M-dimensional concept probabilities is identical for all methods.

For the clinical datasets, we also use architectures tailored for learning checklists, namely Unit weighting, SETS checklist, Integer Linear Program (Zhang et al., 2021) with mean thresholds, and MIP from Makhija et al. (2022). These checklist-specific architectures lack the ability to process complex data modalities such as time series data, so we use features mean values for training them.

**Unit weighting** distils a pre-trained logistic regression into a checklist, as also used in Zhang et al. (2021). First, we binarise the data by using the mean feature values as thresholds. By training a simple logistic regression on the continuous data, we can determine the features with negative weights and replace them with their complements. A hyperparameter $\beta$ to discard features with weights in the range $[-\beta, \beta]$. Subsequently, we train logistic regression models on the binarized data by varying $T \in [0, M]$ (the checklist threshold) to identify the optimal value of T.

**SETS checklist** (Makhija et al., 2022) is an improvement over Unit Weighting, where mean values are directly utilized for binarization. This method aims to learn feature thresholds ($\mu$) while training a logistic regression in a temperature parameter ($\tau$) followed by unit weighting to generate the checklist.

$$\sigma(\frac{X - \mu}{\tau}) \quad \text{where,} \ \sigma \ \text{sigmoid function.} \tag{5}$$

For **Integer Linear Program with Mean Thresholds**, we re-implement the ILP from Zhang et al. (2021) on the mean binarized data and solve the optimization problem using Gurobi. Comparing our method to this baseline provides the most relevant benchmark for evaluation.

**Mixed Integer Program** (Makhija et al., 2022) is a modification of the ILP described above and contains additional constraints to learn thresholds instead of using fixed ones. Again, we re-implement the optimization problem and solve it on Gurobi.

**BERT/LSTM/CNN + LR** employs the concept learners used in ProbChecklist followed by a logistic regression layer on the combination of last layer's embeddings for each modality. Finally, cross-entropy loss is calculated on the predicted class, i.e. $\mathbb{P}(\hat{\mathbf{y}} = \mathbf{1})$.

### E.2 Reasons for creating MNIST synthetic Dataset

Since there aren't any real-world datasets with a ground truth checklist, we first validate our idea on a synthetic setup where the checklist is known. The key points we examine were the performance of our probabilistic checklist objective compared to standard MLP and LR architectures, the comprehensibility of the concepts learnt, and how well our approach recovers the defined rules.

The ideal dataset needed to substantiate our approach comprises different features (or modalities) and a defined rule/condition for each modality to assign the samples a ground truth label. Similar to the structure of a checklist, the sample label depends on how many rules.

These points were taken into account while designing the MNIST Synthetic Checklist.

### E.3 Datasets

**MNIST Checklist Dataset**
We generate a synthetic dataset from MNIST and define a rule set based on which the instances are classified as 0 or 1. We divide all the images in the MNIST dataset into sequences of $\mathbf{K}$ images, and each sequence forms one sample. In the experiments, we set $\mathbf{K} = \mathbf{4}$, which creates a dataset consisting of 10500 samples for training, 4500 for validation, and 2500 for testing. We start by learning $\mathbf{M}$ concepts from images directly using $\mathbf{K}$ concepts learners, which will later be used for recovering the set checklist. Concept learners would ensure that additional information beyond the labels is also captured.

We construct a ground truth checklist using the image labels to simulate a binary classification setup. A sample is assigned $\mathbf{y} = +\mathbf{1}$ if $\mathbf{T}$ out of $\mathbf{K}$ items are true. The final checklist used for experimentation is:

- **Ground Truth Checklist ($\mathbf{T} = 3$, $\mathbf{K} = 4$)**
    - ☐ **Image 1** $\in \{0, 2, 4, 6, 8\}$
    - ☐ **Image 2** $\in \{1, 3, 5, 7, 9\}$
    - ☐ **Image 3** $\in \{4, 5, 6\}$
    - ☐ **Image 4** $\in \{6, 7, 8, 9\}$

The final dataset contains 20.44% positive samples.

**PhysioNet Sepsis Tabular Dataset**
We use the PhysioNet 2019 Early Sepsis Prediction time series dataset (Reyna et al., 2019), which was collected from the ICUs of three hospitals. Hourly data of vital signs and laboratory tests are available for the thirty-four non-static features in the dataset. Additionally, four static parameters, gender, duration, age, and anomaly start point, are included. We treat the occurrence of sepsis in patients as the binary outcome variable.

The original dataset contains 32,268 patients, with only 8% sepsis patients. We create five subsets with 2200 patients, of which nearly 37% are positive. For this task, we use basic summary extraction functions such as the mean, standard deviation, and last entry of the clinical time series of each patient to tabulate the dataset. We subsequently perform feature selection and only keep the top ten informative features ($\mathbf{K} = \mathbf{10}$) based on logistic regression weights.

**PhysioNet Sepsis Time Series Data**
We use the PhysioNet 2019 Early Sepsis Prediction (Reyna et al., 2019) for this task also. The preprocessing steps and formation of subsets are the same as described above. Instead of computing summary statistics from the clinical time series, we train different concept learners to capture the dynamics of the time series. This allows us to encapsulate additional information, such as sudden rise or fall in feature values as concepts.

The processed dataset contains 2272 training, 256 validation, 256 testing samples, and 56 time steps for each patient. We fix $\mathbf{K} = \mathbf{10}$, i.e. use the top ten out of thirty-four features based on logistic

regression weights. The selected features are {temperature, TroponinI, FiO2, WBC, HCO3, SaO2, Calcium, HR, Fibrinogen, AST}.

**MIMIC-III Mortality Time Series Data**
We use the clinical time series data from the MIMIC III Clinical Database (Johnson et al., 2016) to perform mortality prediction on the data collected at the intensive care facilities in Boston. We directly use the famous MIMIC-Extract (Wang et al., 2020) preprocessing and extraction pipeline to transform the raw Electronic Medical Records consisting of vital signs and laboratory records of more than 50,000 into a usable time series format. The processed dataset had a severe class imbalance with respect to the mortality prediction task. We randomly sample from the negative class to increase the percentage to 25% positive patients. Subsequently, we divide the samples into training, validation, and test subsets comprising 6912, 768, and 2048 patients. Like the PhysioNet experiments, we retain the top ten time-series features ($K = 10$). These features are {heart rate, mean blood pressure, diastolic blood pressure, oxygen saturation, respiratory rate, glucose, blood urea nitrogen, white blood cell count, temperature, creatinine}. To evaluate the proposed fairness regularizer, we introduce the one-hot encodings of two categorical features, gender and ethnicity.

**Medical Abstracts TC Corpus.** We work with the clinical notes dataset designed for multi-class disease classification (Schopf et al., 2023). However, we only focus on neoplasm detection. This subset contains 14438 total samples consisting of 11550 training samples and 2888 testing samples. Out of these, 2530 were positive in training set and 633 were positive in the testing set. To reduce class imbalance of 21.9%, the negative set was subsampled to result in an ratio of positive samples to negative samples of 35%. We use medical abstracts which describe the conditions of patients. Each note is 5-6 sentences long on average. We consider the whole text as one modality ($K = 1$).

### E.4 SEPSIS PREDICTION USING STATIC TABULAR DATA

**Data.** We use the PhysioNet 2019 Early Sepsis Prediction time series dataset (Reyna et al., 2019). We transformed this dataset into tabular data by using basic summary extraction functions such as the mean, standard deviation, and last entry of the clinical time series of each patient. We subsequently perform feature selection and only keep the top ten informative features ($K = 10$) based on logistic regression weights.

**Architecture.** Since each feature is simply a single-valued function ($x_i \in \mathbb{R}$), the concept learners are single linear layers followed by a sigmoid activation function. The remaining steps are the same as the previous task, i.e. these concept probabilities are then passed to the probabilistic logic module for $\mathbb{P}(d > T)$ computation and loss backpropagation.

**Baselines.** We use standard baselines like MLP and LR, along with checklist-specific architectures, namely Unit weighting, SETS checklist, Integer Linear Program (Zhang et al., 2021) with mean thresholds, MIP from Makhija et al. (2022). Unit weighting distils a pre-trained logistic regression into a checklist, as also used in Zhang et al. (2021). SETS checklist (Makhija et al., 2022) consists of a modified logistic regression incorporating a temperature parameter to learn feature thresholds for binarization, this is followed by unit weighting.

**Results.** We present the results of ProbChecklist and baselines in Table 4. Our performance is slightly lower than the MIP baseline. The highest accuracy is achieved by an MLP, but that comes at the cost of lower interpretability.

| Model | Accuracy | Precision | Recall | Specificity | M | T |
|---|---|---|---|---|---|---|
| Dummy Classifier | 37.226 | 0.372 | 1 | 0.628 | - | - |
| MLP Classifier | **64.962 ± 2.586** | 0.5726 ± 0.046 | 0.483 ± 0.074 | 0.76043 ± 0.0562 | - | - |
| Logistic Regression | 62.555 ± 1.648 | 0.624 ± 0.0461 | 0.144 ± 0.0393 | **0.9395 ± 0.0283** | - | - |
| Unit Weighting | 58.278 ± 3.580 | 0.521 ± 0.093 | 0.4386 ± 0.297 | 0.6861 ± 0.251 | 9.6 ± 0.8 | 3.2 ± 1.16 |
| SETS Checklist | 56.475 ± 7.876 | 0.517 ± 0.106 | **0.6639 ± 0.304** | 0.494 ± 0.3195 | 10 ± 0 | 6 ± 0.632 |
| ILP mean thresholds | 62.992 ± 0.82 | 0.544 ± 0.087 | 0.1196 ± 0.096 | 0.9326 ± 0.0623 | 4.4 ± 1.01 | 2.8 ± 0.748 |
| MIP | 63.688 ± 2.437 | 0.563 ± 0.050 | 0.403 ± 0.082 | 0.7918 ± 0.06 | 8 ± 1.095 | 3.6 ± 0.8 |
| Concepts + LR | 61.168 ± 1.45 | 0.565 ± 0.059 | 0.324± 0.15 | 0.805 ± 0.1 | - | - |
| **ProbChecklist** | 62.579 ± 2.58 | **0.61 ± 0.076** | 0.345 ± 0.316 | 0.815 ± 0.185S | 10 | 3.6 ± 1.2 |

Table 4: Performance results for Sepsis Prediction task using Tabular Data.

### E.5 SEPSIS PREDICTION USING TIME SERIES DATA

Instead of computing summary statistics from the clinical time series, we train different concept learners to capture the dynamics of the time series. This encapsulates additional information, such as sudden rise or fall in feature values as concepts. Like the setup for tabular dataset, we fix $\mathbf{K} = \mathbf{10}$, i.e. use the top ten features based on logistic regression weights.

We define $\mathbf{K}$ CNNs with two convolutional layers which accept one-dimensional signals as the concept learners for this task. We summarise the results for this task in Table 5. We find that ProbChecklist improves upon the baselines herein as well.

| Model | Accuracy | Precision | Recall | Specificity | $d'_k$ | M | T |
|---|---|---|---|---|---|---|---|
| Logistic Regression | 60.627 ± 1.379 | 0.4887 ± 0.106 | 0.1843 ± 0.073 | 0.8792 ± 0.048 | | - | - |
| Unit Weighting | 60.532 ± 1.567 | 0.4884 ± 0.087 | 0.1882 ± 0.102 | 0.8745 ± 0.066 | | 5 ± 0.63 | 9.2 ± 0.748 |
| ILP mean thresholds | 62.481 ± 0.426 | 0.529 ± 0.242 | 0.0529 ± 0.051 | 0.964 ± 0.031 | 1 | 3.4 ± 1.496 | 3.6 ± 1.85 |
| MIP Checklist | 60.767 ± 1.022 | 0.5117 ± 0.055 | 0.142 ± 0.05 | **0.912 ± 0.036** | | 3.5 ± 0.866 | 6.5 ± 1.5 |
| CNN + MLP | 63.465 ± 2.048 | 0.585 ± 0.05 | 0.234 ± 0.071 | 0.895 ± 0.021 | - | | - |
| CNN + MLP (all features) | **67.19 ± 0.32** | 0.526 ± 0.065 | 0.281 ± 0.041 | 0.835 ± 0.033 | | - | - |
| **ProbChecklist** | **63.671 ± 1.832** | **0.609 ± 0.115** | **0.354 ± 0.157** | 0.823 ± 0.108 | 3 | 4.4 ± 1.356 | 30 |

Table 5: Performance results for Sepsis Prediction task using Tabular Data.

### E.6 SENSITIVITY ANALYSIS

We study the sensitivity of our approach to the number of learnable concepts. In particular, we compare the performance of ProbChecklistwith increasing number of learnt concepts ($d'_k$). The observed trend for both MNIST (Table 6) and PhysioNet (Table 9) datasets exhibited similarities. We noted a significant improvement in accuracy, precision, and recall when $d'_k$ increased from 1 to 2, suggesting that the samples contain complex features that a single concept cannot represent. Once all the valuable information is captured, accuracy reaches a saturation point and the performance plateaus with $d'_k$.

| $d'_k$ | Evaluation | Accuracy | Precision | Recall | Specificity | T | M |
|---|---|---|---|---|---|---|---|
| 1 | Model | 86.04 ± 0.463 | 0.814 ± 0.027 | 0.412 ± 0.021 | 0.976 ± 0.004 | 3 | 4 |
| | Checklist | 72.63 ± 1.42 | 0.427 ± 0.013 | **0.979 ± 0.005** | 0.661 ± 0.018 | | |
| 2 | Model | 96.888 ± 0.064 | 0.925 ± 0.008 | **0.922 ± 0.012** | 0.981 ± 0.003 | 5 | 8 |
| | Checklist | 92.768 ± 1.6 | 0.752 ± 0.045 | 0.97 ± 0.006 | 0.917 ± 0.02 | | |
| 3 | Model | **97.064 ± 0.187** | 0.94 ± 0.008 | 0.915 ± 0.013 | 0.985 ± 0.002 | 7 | 12 |
| | Checklist | 96.52 ± 0.33 | 0.9 ± 0.009 | 0.933 ± 0.008 | 0.973 ± 0.002 | | |
| 4 | Model | 97.04 ± 0.177 | 0.937 ± 0.005 | 0.917 ± 0.006 | 0.984 ± 0.001 | 8.4 ± 1.2 | 16 |
| | Checklist | **96.808 ± 0.24** | 0.917 ± 0.015 | 0.929 ± 0.01 | 0.978 ± 0.004 | | |
| 5 | Model | 97.032 ± 0.135 | **0.943 ± 0.005** | 0.91 ± 0.01 | **0.986 ± 0.001** | 9.4 ± 1.36 | 20 |
| | Checklist | 96.68 ± 0.269 | **0.918 ± 0.013** | 0.92 ± 0.009 | **0.979 ± 0.004** | | |

Table 6: Performance of ProbChecklist with varying $\mathbf{d'_k}$ on MNIST Checklist Dataset

### E.7 CHECKLIST OPTIMIZATION

The checklist generation step from the learnt concepts allows users to tune between sensitivity-specificity depending on the application. The optimal checklist can be obtained by varying the threshold ($\tau$) used to binarize the learnt concepts ($\mathbf{c}_i[j] = \mathbb{I}[\mathbf{p}_i[j] > \tau]$) to optimize the desired metric (Accuracy, F1-Score, AUC-ROC) on the validation data. In Tables 8 and 7, we compare the performance of the checklist obtained by varying the optimization metric.

In the case of the MIMIC mortality prediction task, we also train our models using Binary Cross Entropy (BCE) loss and optimize over the threshold used to binarize the prediction probabilities. Next, we perform a pairwise comparison between the binary cross-entropy loss and ProbChecklist

loss for each optimization metric. From the results in Table 7, it is evident that our method achieves superior performance in terms of accuracy for both metrics.

| Loss | Checklist Optimization | Accuracy | Precision | Recall | Specificity | T | M |
|------|------------------------|----------|-----------|--------|-------------|---|---|
| BCE Checklist | Accuracy | 76.4±0.6
**76.61±0.52** | **0.61±0.03**
0.59±0.05 | 0.22±0.09
**0.24±0.06** | **0.95±0.02**
0.94±0.02 | -
6.6±1.74 | -
20 |
| BCE Checklist | F1-Score | 67.7±3.04
**71.45±1.66** | 0.41±0.03
**0.45±0.02** | **0.62±0.1**
0.58±0.04 | 0.7±0.07
**0.76±0.04** | -
7.4±1.36 | -
20 |
| Checklist | AUC-ROC | 34.49±4.18 | 0.27±0.01 | 0.98±0.01 | 0.13±0.06 | 4±0 | 20 |

Table 7: Results for different Checklist Optimization methods (Accuracy, F1-Score, AUC-ROC) on the MIMIC Mortality Prediction Task for $d'_k = 2$. We report results for the same architecture trained using Binary Cross Entropy (BCE) loss (instead of the proposed ProbChecklist loss) for a fair comparison.

For the PhysioNet Sepsis Prediction Task, we analyze a different aspect of our method. We study the difference in performance by changing the number of learnt concepts per modality ($d'_k$) for all the metrics.

| $d'_k$ | Checklist Optimization | Accuracy | Precision | Recall | Specificity | T | M |
|--------|------------------------|----------|-----------|--------|-------------|---|---|
| 1 | Accuracy
F1-Score
AUC-ROC | 62.5±1.5
49.22±9.74
40.72±3.72 | 0.67±0.18
0.44±0.06
0.39±0.01 | 0.21±0.13
0.85±0.11
0.93±0.12 | 0.9±0.1
0.27±0.22
0.08±0.13 | 4±0.89
3.8±0.98
4±1.22 | 10 |
| 2 | Accuracy
F1-Score
AUC-ROC | 62.97±3.36
38.98±1.9
48.34±10.56 | 0.58±0.06
0.39±0.02
0.29±0.17 | 0.34±0.18
0.98±0.02
0.54±0.41 | 0.82±0.15
0.01±0.01
0.43±0.43 | 3±1.1
2.6±0.8
3.75±1.09 | 20 |

Table 8: Results for different Checklist Optimization methods (Accuracy, F1-Score, AUC-ROC) on the PhysioNet Sepsis Prediction Task using timeseries. We report results for different values of $d'_k$.

### E.8 IMPACT OF THE BINARIZATION SCHEME

In Tables 9 and 6 we show the difference in performance between two types of evaluation schemes. The first scheme is the typical checklist, obtained by binarizing the concept probabilities and thresholding the total number of positive concepts at $\mathbf{T}$. The second is the direct model evaluation, using the non-binarized output probability. It involves only thresholding $\mathbb{P}(\mathbf{d} > \mathbf{T})$ to obtain predictions. As expected, the performance of the checklist is slightly lower than the direct model evaluation because concept weights are binarized leading to a more coarse-grained evaluation.

| $d'_k$ | Evaluation | Accuracy | Precision | Recall | Specificity | T | M |
|--------|------------|----------|-----------|--------|-------------|---|---|
| 1 | Model
Checklist | 64.105 ± 1.69
63.716 ± 3.02 | 0.586 ± 0.047
0.613 ± 0.12 | 0.309 ± 0.025
0.233 ± 0.035 | 0.857 ± 0.018
0.9 ± 0.036 | 3 ± 0.89 | 10 |
| 2 | Model
Checklist | 62.656 ± 2.377
62.969 ± 3.355 | 0.525 ± 0.025
0.582 ± 0.061 | 0.565 ± 0.032
0.343 ± 0.176 | 0.667 ± 0.027
0.817 ± 0.145 | 3 ± 1.095 | 20 |
| 3 | Model
Checklist | 60.781 ± 2.09
63.671 ± 1.832 | 0.503 ± 0.019
0.609 ± 0.115 | 0.545 ± 0.026
0.354 ± 0.157 | 0.648 ± 0.019
0.823 ± 0.108 | 4.4 ± 1.356 | 30 |

Table 9: Evaluation of the binarization scheme for sepsis prediction task using time series data.

## F CONCEPTS INTERPRETATION

This section provides a detailed analysis of the learnt concepts using TANGOS regularization outlined in Section 4.6.

## F.1 TANGOS REGULARIZATION

TANGOS regularization assists in quantifying the contribution of each input dimension to a particular concept. We examine the gradient of each concept obtained from the concept extractors with respect to the input signal. The TANGOS loss consists of two components: The first component enforces sparsity, emphasizing a concentrated subset of the input vector for each concept. The second component promotes uniqueness, minimizing the overlap between the input subsets from which each concept is derived. Sparsity is achieved by taking the L1-norm of the concept gradient attributions with respect to the input vector. To promote decorrelation of signal learned in each concept, the loss is augmented by incorporating the inner product of the gradient attributions for all pairs of concepts.

The degree of interpretability of the concepts can be varied by changing the regularization weights in the TANGOS loss equation 6. For a stronger regularization scheme (i.e. higher regularization weights), the gradient attributions are disentangled and sparse.

$$\mathcal{L}_{TANGOS} = \lambda_{sparsity}\mathcal{L}_{sparsity} + \lambda_{correlation}\mathcal{L}_{correlation} \tag{6}$$

$$= \frac{\lambda_{sparsity}}{N} \sum_{n=1}^{N} \frac{1}{d'_k} \sum_{j=1}^{d'_k} ||\frac{\partial p_j(x)}{\partial x_i}||_1 \tag{7}$$

$$+ \frac{\lambda_{correlation}}{N} \sum_{n=1}^{N} \frac{1}{d'_k C_2} \sum_{j=2}^{d'_k} \sum_{l=1}^{d'_k-1} \frac{\langle a^j(x_i), a^l(x_i) \rangle}{||a^j(x_i)||_2, ||a^l(x_i)||_2} \tag{8}$$

where, $a^j(x_i) = \dfrac{\partial p_j(x)}{\partial x_i}$ represents the attribution of $j^{th}$ concept with respect to the $i^{th}$ patient ($x_i$).

To supplement convergence and enhance the stability of our models, we resort to annealing techniques to gradually increase the weights of these regularization terms.

## F.2 MNIST IMAGES

We consider the experimental setup where each encoder learns two concepts per image. Figure 5 shows the gradient attribution heatmaps of Image 2 of the MNIST Dataset with respect to both concepts for four cases. The ground truth concept for this image is **Image 2** $\in \{1, 3, 5, 7, 9\}$.

We observe that there is a significant overlap among the gradient attributions when no regularisation terms were used ($\lambda_{sparsity} = 0$ and $\lambda_{correlation} = 0$), indicating redundancy in the learnt concepts (almost identical representations). However, it manages to identify odd digits correctly and performs well.

For $\lambda_{sparsity} = 10$ and $\lambda_{correlation} = 1$, the concepts correspond to simple features like curvatures and straight lines and are easy to identify. We infer from the gradient heat maps that concepts 1 and 2 focus on the image's upper half and centre region, respectively. Concept 1 is true for digits 5, 8, 9 and 7, indicating that it corresponds to a horizontal line or slight curvature in the upper half. Since Digits 0 and 2 have deeper curvature than the other images, and there is no activity in that region in the case of 4, concept 1 is false for them. concept 2 is true for images with a vertical line, including digits 9, 4, 5 and 7. Therefore, concept 2 is false for the remaining digits (0, 2, 8).

Table 10 highlights the tradeoff between interpretability of the concepts and the checklist performance. By increasing regularization weights, even though checklist accuracy decreases, the gradient attributions disentangle and become sparse. Based on our experiments, the checklist performance is more sensitive to the sparsity weight (i.e. decreases more sharply).

We extend this analysis to the setting with four learnable concepts $d'_k = 4$. Figure 6 ($\lambda_{sparsity} = 10$ and $\lambda_{correlation} = 1$) shows that each concept is focusing on different regions of the image. At least three concepts out of four are true for odd digits, whereas not more than one sample is true for negative samples. This attests to the ability of the method to learn underlying concepts correctly.

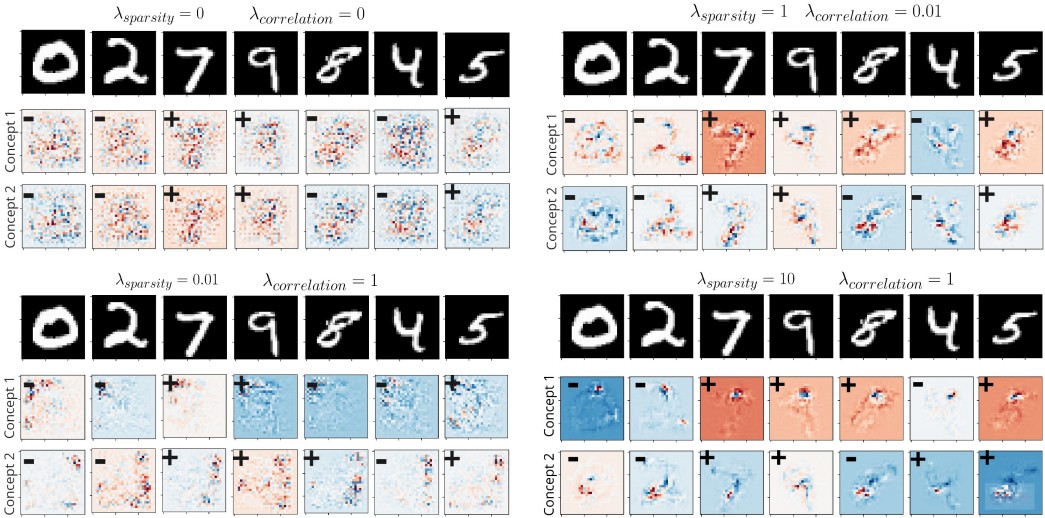

Figure 5: We plot images and corresponding gradient attributions heat maps for seven input samples of the Image 2 modality of the MNIST dataset for different combinations of sparsity and correlation regularization terms. We used a checklist with two learnable concepts per image. The intensity of red denotes the positive contribution of each pixel, whereas blue indicates the negative. If a concept is predicted as true for an image, then we represent that with a plus (+) sign and a negative (-) otherwise.

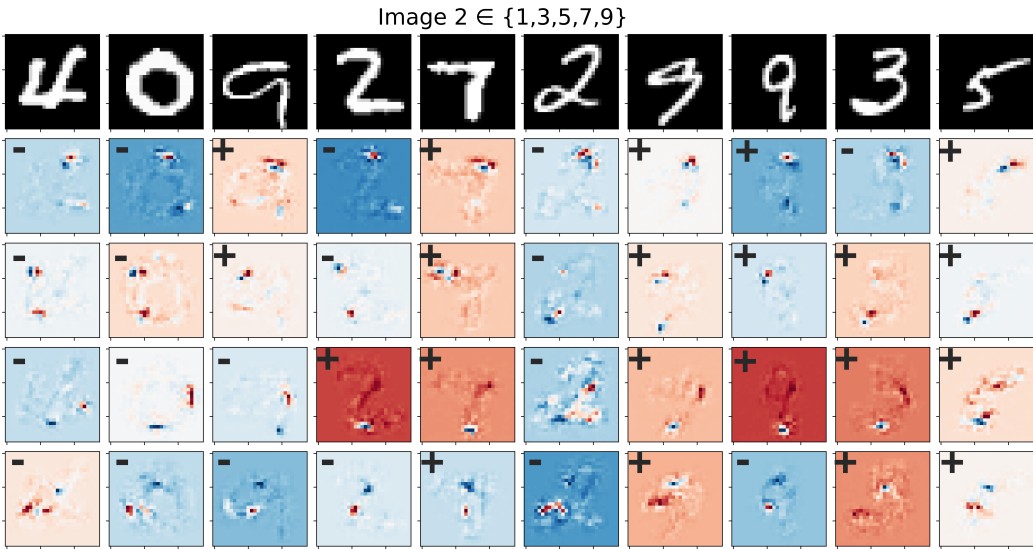

Figure 6: We plot images and corresponding gradient attributions heat maps for seven input samples of the Image 2 modality of the MNIST dataset for $\lambda_{sparsity} = 10$ and $\lambda_{correlation} = 1$. We used a checklist with four learnable concepts per image. The intensity of red denotes the positive contribution of each pixel, whereas blue indicates the negative. If a concept is predicted as true for an image, then we represent that with a plus (+) sign and a negative (-) otherwise.

| $d'_k$ | $\lambda_{\text{sparsity}}$ | $\lambda_{\text{correlation}}$ | **Accuracy** | **Precision** | **Recall** | **Specificity** |
|---|---|---|---|---|---|---|
|   | 0 | 0 | 95.48 | 0.829 | 0.981 | 0.948 |
|   | 1 | 0.01 | 93.40 | 0.761 | 0.988 | 0.920 |
| 2 | 0.01 | 1 | 92.88 | 0.745 | 0.990 | 0.913 |
|   | 10 | 1 | 79.28 | 0.485 | 0.978 | 0.733 |
|   | 0 | 0 | 97.20 | 0.933 | 0.930 | 0.983 |
|   | 1 | 0.01 | 97.04 | 0.901 | 0.961 | 0.973 |
| 4 | 0.01 | 1 | 96.96 | 0.904 | 0.953 | 0.974 |
|   | 10 | 1 | 78.88 | 0.491 | 0.844 | 0.775 |

Table 10: Performance of ProbChecklist for different combinations of sparsity ($\lambda_{sparsity}$) and correlation ($\lambda_{correlation}$) regularization weights on MNIST Checklist Dataset

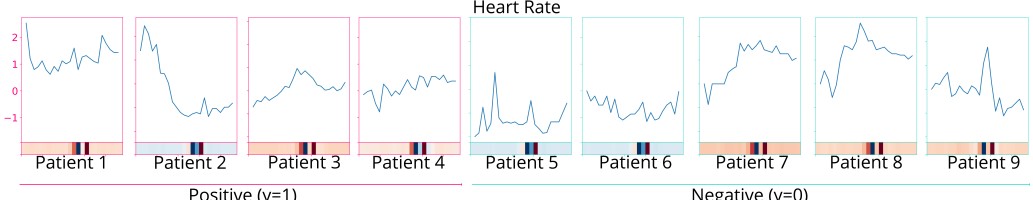

Figure 7: We plot time series for Heart Rate and corresponding gradient attributions for nine patients at $\lambda_{sparsity} = 0.1$ and $\lambda_{correlation} = 10$. The intensity of red denotes the positive contribution of that time step, whereas blue indicates the negative. The border colour of each sample encodes the ground-truth label, with pink being the positive outcome (i.e. mortality).

### F.3 MIMIC TIME SERIES

For this analysis, we again fix our training setting to $d'_k = 2$. Sudden changes in slope in the clinical features were associated with switching of the sign gradient attributes. In Figure 7, we plot the time series for heart rate and corresponding gradient attributions for the first concept learnt by CNN. Maximum activity is observed between time steps 12 to 17 for all the patients. If a positive peak (or global maxima) is observed in this period (patients 3, 6, 7, 8), then the gradient at all other time steps has a positive contribution. Another interesting observation is that if the nature of the curve is decreasing and the significant portion has negative values (Images 2, 5, 6), then the surrounding gradient attributions have a negative impact. These features are consistent irrespective of the patient outcome.

From Figure 8, we infer that gradient attributions for both the concepts are identical. In the absence of regularization terms, it was very hard to explain the concepts that were being learnt. Even though interpretability comes at the cost of performance, it helped in mapping learnt concepts to archetypal signals and patterns in the timeseries. Sudden changes in slope in the clinical features were associated with switching of the sign gradient attributes.

With the help of Figure 9, we infer the concepts being learnt when the regularization weights are $\lambda_{sparsity} = 0.1$ and $\lambda_{correlation} = 10$. The region between time steps 11 to 16 is activated for Heart Rate. The learnt concept is negative when there is a sudden increase (peak) in the values around this region as observed in Patients $4, 5, 6, 7$. This concept is true when there is slight fluctuation, or the values are steady around the high gradient region, as seen in the remaining patients. For White Blood Cell Count, our model learns two distinct concepts. The neuron gradient attributions for the first concept are high for time steps 3 to 8. This concept is positive when a local maxima is observed (an increase from the starting value and then a decrease). This pattern is visible in Patients $2, 5$. In all the other cases, a local minima is observed first.

### F.4 MEDICAL ABSTRACTS CORPUS

Since we only work with text data, we assumed that the concepts learnt are functions of tokens in the text. To obtain interpretable summaries of the concept, we train decision trees on token occurrence

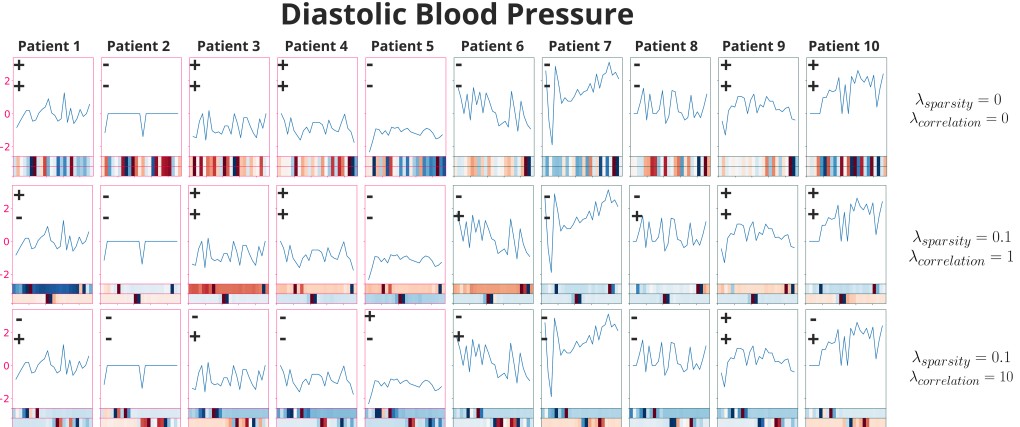

Figure 8: We plot time series for Diastolic Blood Pressure and corresponding gradient attributions for for both concepts for different combinations of regularization weights. The intensity of red denotes the positive contribution of that time step, whereas blue indicates the negative. The border colour of each sample encodes the ground-truth label, with pink being the positive outcome (i.e. mortality). If a concept is predicted as true for a patient, then we represent that with a plus (+) sign and a negative (-) otherwise.

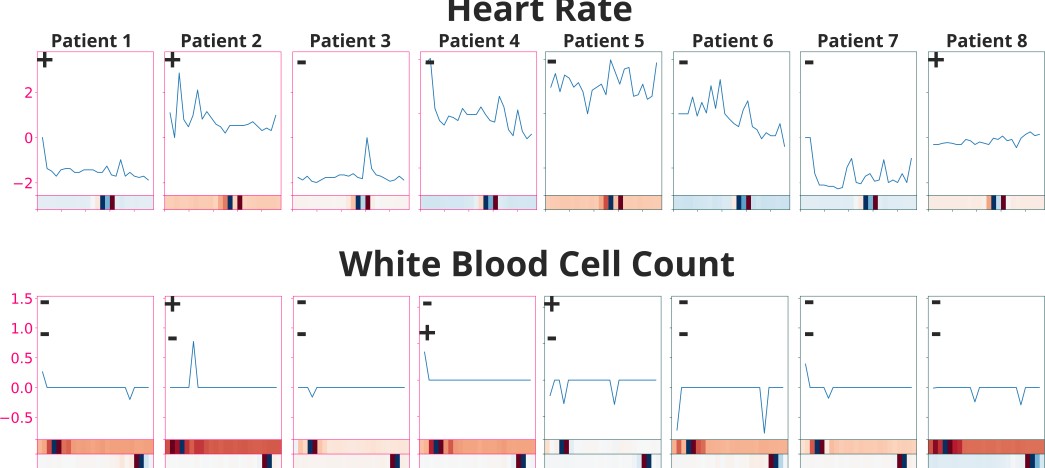

Figure 9: We plot time series for Heart Rate and White Blood Cell Count and corresponding gradient attributions for eight patients for $\lambda_{sparsity} = 0.1$ and $\lambda_{correlation} = 10$. The intensity of red denotes the positive contribution of that time step, whereas blue indicates the negative. The border colour of each sample encodes the ground-truth label, with pink being the positive outcome (i.e. mortality). If a concept is predicted as true for a patient, then we represent that with a plus (+) sign and a negative (-) otherwise.

and use the concept predictions as labels. We then represent each concept by the list of tokens used in the five first layers of each decision tree. We prune this representation by only keeping the unique tokens for each concept.

# G    ADDITIONAL FAIRNESS RESULTS

Additionally, we provide FNR and FPR values for each subgroup both before and after applying regularization (Table 11). This is done to ensure that the error rates of majority subgroups, where the performance was originally strong, do not increase. Notably, most minority subgroups benefit from this regularization; however, FNR increases for both the Female (minority) and Male (majority) subgroups after regularization.

| Subgroup | Before/After FR | FPR | FNR |
|---|---|---|---|
| **Female** | Before | 0.719 | 0.0989 |
| | After | 0.1899 | 0.6975 |
| **Male** | Before | 0.4969 | 0.2931 |
| | After | 0.127 | 0.851 |
| **White** | Before | 0.782 | 0.937 |
| | After | 0.072 | 0.0868 |
| **Black** | Before | 0.661 | 0.824 |
| | After | 0.0563 | 0.0597 |
| **Others** | Before | 0.724 | 0.913 |
| | After | 0.064 | 0.0805 |

Table 11: We provide the actual values of FNR and FPR for each subgroup before and after fairness regularization is applied.

| | Accuracy | Recall | Precision |
|---|---|---|---|
| Without Fairness Regularizer | 75.99 | 0.117 | 0.648 |
| With Fairness Regularizer | 72.412 | 0.256 | 0.553 |

Table 12: We present accuracy, precision, and recall metrics to compare the model's performance before and after applying fairness regularization. Despite observing a decrease in accuracy, the increase in recall signifies a positive development.

# H    EXTENSION OF PROBCHECKLIST: CHECKLISTS WITH LEARNABLE INTEGER WEIGHTS

Checklist structure considers the weight of each item on the checklist as +1. In this section, we propose an extension to allow for integer weights larger than 1 because it has many useful applications and may be of interest to users. Before we formulate a method to learn integer weights for each concept, it is important to note that this would make it harder to interpret the checklist. More specifically, the meaning of the checklist threshold used for classification of samples would now change. Previously, it represented the minimum number of true concepts for a positive classification. Now it signifies a score - the minimum value of the weighted sum of concept probabilities for a positive classification.

Given K data modalities as the input for sample i, we train K concept learners to obtain the vector of probabilistic concepts of each modality $\mathbf{p_i^k} \in [0, 1]^{d'_k}$. Next, we concatenate the full concepts probabilities ($\mathbf{p_i}$) for sample i. At this point, we can introduce a trainable weight vector $W \in [0, 1]^{d'}$ (with $\sum_{i=1}^{d'} w_i = 1$) of the same dimension as $p_i$ (concept probabilities) which will capture the

relative importance of the features. Element-wise product ($\circledcirc$) of $p_i$ and W represents the weighted concept probabilities and can be denoted with $Wp_i$. This vector can be normalized by dividing each element with the maximum entry (L0 norm). While training it i For training the concept learners, we pass $Wp_i$ through the probabilistic logic module. After training, the integer weights corresponding to each concept in the checklist can be obtained by converting the W to a percentage: $W_{int}$. At inference time, we discretize $\mathbf{C_i}$ to construct a complete predictive checklist. Next, compute the score $W_{int}^T C_i$ and compare it against the checklist threshold, $M$, to classify the sample.

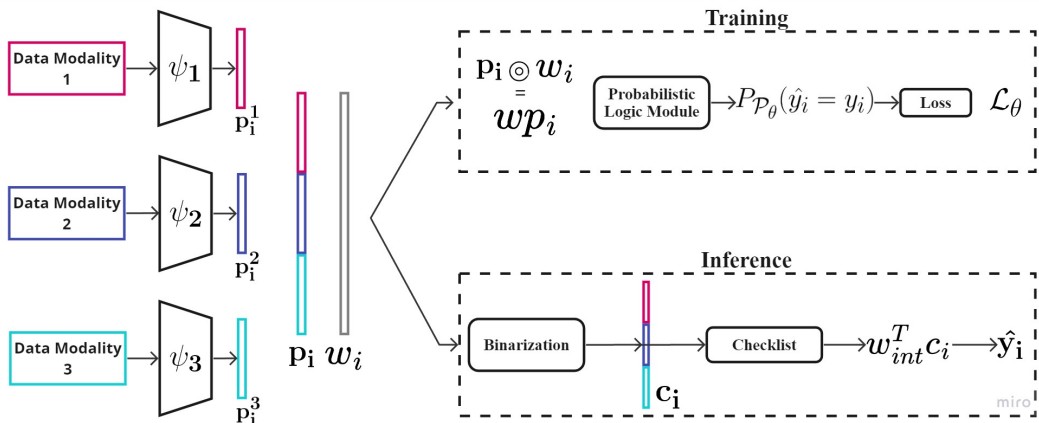

## I PLOT: PERFORMANCE RESULTS OF PROBCHECKLIST

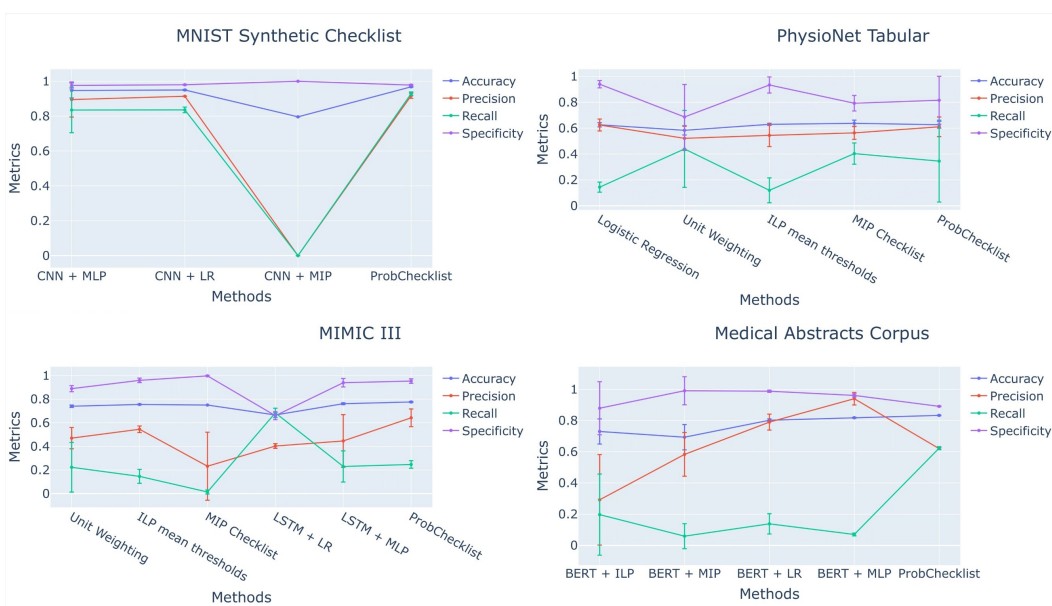

Figure 10: We plot the performance results reported in Table 1.

## J IMPLEMENTATION

The code for all the experiments and instructions to reproduce the results are available at `https://anonymous.4open.science/r/ProbChecklist-322A/`. We have extensively used the PyTorch(Paszke et al., 2019) library in our implementations and would like to thank the authors and developers.