# OpenReview forum: "Learning Predictive Checklists with Probabilistic Logic Programming"
_ICLR.cc/2024/Conference — Submitted to ICLR 2024_

### Official Review · Reviewer_SxKf · 2023-10-30

**Soundness:** 3 good
**Presentation:** 2 fair
**Contribution:** 3 good
**Rating:** 8
**Confidence:** 4

**Summary:**

The authors propose a new framework for learning predictive checklists. The method is able to process time series, images, tabular features, etc. The use of techniques enabling sparse representations of the inputs and the use of a fairness metric lead to checklists that would have interpretable features and that promote fairness toward sensible variables.

**Strengths:**

-The « related works »  analysis is thorough and seems up-to-date.
-I found the experiment subsection 5.1 truly compelling. Many metrics are reported, which I think is not done often enough.
-The approach is well-explained and flexible.

**Weaknesses:**

**Major**

1 – Most how the points I would like to raise concern interpretability. (See also the points in the Question section on that matter.)

1.1 - Interpretability is directly impacted by the complexity of the model itself, but the fact that the algorithm is in itself a black box makes it such that understanding why the model is what it is is unreachable.

1.2 – As discussed in [1], p.17, when it comes to logical rules, the more digits there are to take into account, the less the rule is interpretable. One could argue that the checks from Figure 4 aren’t that interpretable. When it comes to the features themselves: what does it really mean to have an « sd » of FiO2 above 0.035? It is normal? Is it higher than the average? Is it higher than a certain minimum threshold? Such questions arrises with every check.

1.3 – The interpretation made of the checks from MNIST Images task is questionable. It’s been known for a long time now that saliency maps, especially the vanilla approach of looking at the gradient map, can easily lead to false conclusions, especially when those conclusions match what a person is seeking [2]. Interpretability is needed in the context where explainability (e.g. saliency maps) is not trustworthy.

1.4 – Finally, the interpretation of the features is made in an example where the relationship between the inputs of a problem and the labels is known. The procedure given in order to make sense of the feature extract wouldn’t work if this knowledge was unknown.

2.1 – It is shown that the use of the fairness regularizer works in order to minimize both FNR and FPR, but it is not discussed whether or not the constraint impacts the performances of the checklist, so there is no way to truly understand if its usage is really beneficial.

2.2 – The second contribution states « We investigate the impact of different schemes for improving the interpretability of the concepts learnt as the basis of the checklist. We employ regularization techniques to encourage the concepts to be distinct, so they can span the entire input vector and be specialized, i.e. ignore the noise in the signal and learn sparse representations. We also investigate the impact of incorporating fairness constraints into our architecture. » But since (as discussed in 2.1) there lacks evidence of the soundness of the fairness regularizer, combined with the fact that there is no evidence demonstrating that « regularization techniques encourage the concepts to be distinct, so they can span the entire input vector and be specialized » (as discussed in 1.4), or at least that the « different schemes for improving the interpretability of the concepts learned » truly are responsible for such observations, the soundness of all Contribution 2 can be questioned.

[1] : Rudin, C., Chen, C., Chen, Z., Huang, H., Semenova, L., & Zhong, C. (2021). Interpretable Machine Learning: Fundamental Principles and 10 Grand Challenges. ArXiv. /abs/2103.11251

[2] : Julius Adebayo, Justin Gilmer, Michael Muelly, Ian J. Goodfellow, Moritz Hardt, and Been Kim. 2018. Sanity Checks for Saliency Maps. In NeurIPS. 9525–9536.

**Minor**

1 – Typos. There are several of them...

1.1 - « Figure 1: Example checklist **learnt** by our architecture. Three **of** more checks […]. »

1.2 - « Clinical practice is **an** highly stressful [...] »

1.3 - « […] programming and thus exhibits much faster **?** times and [...] » (a word is missing; « computation », « training »?)

1.4 - « […] we can write the **probabality** of query q as follows. »

1.5 - « We **additional** introduce a regularization [...] »

1.6 - « We investigate the performance **?** ProbChecklist along [...] »

1.7 - « We create **a We** briefly describe the MNIST [...] »

1.8 - « focus on the image’s upper half and **centre** »

1.9 - « we visualize **? learnt** by ProbChecklist in one of the experiments »

1.10 - « Detailed complexity analysis can be found in the **?** B. »

1.11 - « […] interpretable such as decision trees) **)** and posthoc [...] »

2 - « ProbChecklist » is named one time (the first time) before being properly introduced (the second time it is mentioned).

3 – The first citation « Learning Predictive and Interpretable Timeseries Summaries from ICU Data, volume 1, 2021. » doesn’t respect the template. It should be something like « Johnson N, Parbhoo S, Ross AS, Doshi-Velez F. Learning Predictive and Interpretable Timeseries Summaries from ICU Data. AMIA Annu Symp Proc. 2022 Feb 21;2021:581-590. PMID: 35309006; PMCID: PMC8861716. »

4 – The fourth chapter’s title should be isolated from the previous paragraph.

5 – Using a single letter (with the same calligraphy) for two different usages (‘d’, both overall dimension and error criterion) is not desirable.

6 – Constraints are not respected concerning the configuration of the table (Table 1): « number and title always appear before the table » (ICLR24 template and instructions).

7 – In Table 1: why is there no number bolded for some dataset / metric? Why are there two bolded results for MIMIC III – Accuracy?

**Questions:**

1 - Interpretability is not inherent to a family of models. For example, a checklist whose features aren’t interpretable or a checklist with too many « checks » to look at (as with linear models) isn’t interpretable either, for the simple knowledge is drowned in the quantity of information to manipulate. Therefore: how is it made sure that the model, concerning those two criteria, remains interpretable?

2 – It is argued that decision trees could be of lesser interest when it comes to medical applications. But, the interpretation of a model is part of how the features interact with each other in order to generate a given response. When it comes to checklists, no interaction is presented whatsoever; in the case of decision trees, it is inherent what features need to be looked at carefully given the value of some other feature. Wouldn’t that be more appropriate in the context of medical tasks?

3 – It has been briefly discussed that there is an exponential memory complexity intrinsic to the model. Was that a limitation to the experiments that have been run?

4 - How did it impact the training time? Was that training time similar to the compared approaches? How many hyperpameters are there in total, and when compared to the baselines?

**Details Of Ethics Concerns:**

This is not a huge "ethical issue", but the ICLR24 .tex template was modified, most probably in order to save space (no space before and/or after headings in some places, for example, the titles for Sections 4, 5.3, 6; or table titles, see Table 1).

---

> ### Author Response · Authors · 2023-11-20
> **Rebuttal by Authors (1/3)**
>
> We thank the reviewer for their valuable comments and insights.
>
> ```1 - Interpretability is not inherent to a family of models. For example, a checklist whose features aren’t interpretable or a checklist with too many « checks » to look at (as with linear models) isn’t interpretable either, for the simple knowledge is drowned in the quantity of information to manipulate. Therefore: how is it made sure that the model, concerning those two criteria, remains interpretable?```
>
> **Checklist with many items:**
>
> Hyperparameters $d_k, \ T and \ M$ represent the structure/compactness of the checklist. Domain experts will be more equipped to choose these values based on their knowledge of the features. For example, if we are recording a time series feature known to stay stable (not fluctuate much), then low $d_k$ is sufficient. The value of $d_k$ also depends on the number of observations (most blood tests aren’t performed hourly, but heart rate and oxygen saturation are monitored continuously). We encourage the users to select the minimum possible value for $d_k$ to effectively capture all the information. We have performed sensitivity analysis to study the relation between $d_k$ and performance. Figure 3a suggests that the performance saturates after a certain point. This point can be determined experimentally for different modalities. M, total concepts in the checklist, is obtained by pruning concepts that are true for insignificant samples.
>
> **Interpretable Concepts:**
>
> We refer the reviewer to read the discussion on interpretability of concepts in the global response. We would be happy to answer any further questions.
>
> ```2) It is argued that decision trees could be of lesser interest when it comes to medical applications. But, the interpretation of a model is part of how the features interact with each other in order to generate a given response. When it comes to checklists, no interaction is presented whatsoever; in the case of decision trees, it is inherent what features need to be looked at carefully given the value of some other feature. Wouldn’t that be more appropriate in the context of medical tasks?```
>
> Decision trees and checklists are interpretable and widely used in the clinical domain. These models offer different forms of interpretability: checklists provide limited feature interaction, while each path in a decision tree represents a separate checklist.  The main goal of using such models is to automate certain stages of diagnosis/treatment and reduce the burden on the clinicians.
> Checklists, recognized for their robustness, appear as an ideal choice in emergency rooms or scenarios where doctors manage a high volume of patients simultaneously. Their robustness is intrinsically linked to interpretability and can help anticipate hard samples. This avoids poor performance on corner cases, thereby validating the model. We direct the reviewer to Section D of the appendix where we argue the tradeoffs in interpretability between  checklists and decision trees.
>
> ```3 – It has been briefly discussed that there is an exponential memory complexity intrinsic to the model. Was that a limitation to the experiments that have been run?```
>
> We have provided detailed complexity analysis and training details in the appendix (Section B). Yes, the exponential complexity is the primary reason for performing feature selection and limiting the modalities to 10 and learning up to 3 features per modality. A fruitful future direction would be to study approximations to explore a smaller set of combinations.
>
> ```4 - How did it impact the training time? Was that training time similar to the compared approaches? How many hyperpameters are there in total, and when compared to the baselines?```
>
> We thank the reviewer for this suggestion. We have added a comparison of training times with the MIP checklist method on the PhysioNet Tabular dataset to evaluate the extent of this limitation (Section B of the appendix). It's crucial to highlight that while MIP Checklist performs effectively with tabular data, successfully uncovering the optimal solution, its performance is poor when applied to MNIST synthetic setup. Even when we set the runtime for Gurobi solver as 1 hour, it struggles to achieve optimal solutions for many cases. On the other hand, ProbChecklist stands out as more reliable, capable of performing end-to-end training and successfully learning the optimal solution.

---

> ### Author Response · Authors · 2023-11-20
> **Rebuttal by Authors (2/3)**
>
> ```5) Most how the points I would like to raise concern interpretability. (See also the points in the Question section on that matter.)```
>
> ```5.1 - Interpretability is directly impacted by the complexity of the model itself, but the fact that the algorithm is in itself a black box makes it such that understanding why the model is what it is is unreachable.```
>
> We refer the reviewer to our discussion on the interpretability of checklist structure and interpretability of learnt concepts in the global response.
>
> ```5.2 – As discussed in [1], p.17, when it comes to logical rules, the more digits there are to take into account, the less the rule is interpretable. One could argue that the checks from Figure 4 aren’t that interpretable. When it comes to the features themselves: what does it really mean to have an « sd » of FiO2 above 0.035? It is normal? Is it higher than the average? Is it higher than a certain minimum threshold? Such questions arrises with every check.```
>
> We transformed the PhysioNet time series dataset (which contains hourly data) into a tabular format by manually extracting features like mean, standard deviation, and final values. This method of feature extraction is extensively employed in clinical machine learning and is acknowledged as one of the most dependable and interpretable techniques in this domain. The standard deviation is particularly effective in capturing fluctuations in a patient's vital signs, with abrupt changes indicating the necessity for specialized care. We also want to clarify that these concepts are probabilistic i.e. ‘FiO2 > 0.035 with probability 0.5'. We can optimize this probability threshold based on whether the objective is to obtain higher accuracy, AUC-ROC, or F1-Score. We performed experiments by changing the objective, which can be found in Section E.7 of the supplementary material.
>
> ```5.3 – Finally, the interpretation of the features is made in an example where the relationship between the inputs of a problem and the labels is known. The procedure given in order to make sense of the feature extract wouldn’t work if this knowledge was unknown.```
>
> We direct the reviewer to Figure 1 of the paper, which illustrates the checklist learnt from the medical abstract classification task. As previously explained in the global response, understanding results for NLP tasks is significantly simpler due to the comprehensibility of tokens. We have now included a description of the technique used for generating this checklist in the main paper. While experiments involving the TANGOS regularizer on time series datasets have been conducted, we have presented them in the supplementary section. This decision was made as these results require assessment by domain experts to discern patterns effectively.  Instead, our focus has shifted to the synthetic MNIST dataset, as its known ground truth concepts make it easier to validate our approach. We also discuss the results of the PhysioNet Tabular dataset.
>
> ```6) It is shown that the use of the fairness regularizer works in order to minimize both FNR and FPR, but it is not discussed whether or not the constraint impacts the performances of the checklist, so there is no way to truly understand if its usage is really beneficial.```
>
> We agree with the reviewers assessment that only the difference in FNR/FPR error rates for pairs of sensitive groups is insufficient to gauge the fairness regularizer's effectiveness. It is crucial to ensure that the performance of the majority sensitive group does not deteriorate while the minority groups experience improvements. We have reported the FNR/FPR for each sensitive group before and after the regularizer is applied in Section G of the supplementary material. Most minority subgroups benefit
> from this regularization; however, FNR increases for both the Female (minority) and Male (majority)
> subgroups after regularization. We believe that comparing FNR/FPR for individual subgroups is more apt in this setting and provides more fine-grained/thorough evaluation. We have now included additional results on how the model performance varies after the fairness regularizer is applied.

---

> ### Author Response · Authors · 2023-11-20
> **Rebuttal by Authors (3/3)**
>
> ```7) The second contribution states « We investigate the impact of different schemes for improving the interpretability of the concepts learnt as the basis of the checklist. We employ regularization techniques to encourage the concepts to be distinct, so they can span the entire input vector and be specialized, i.e. ignore the noise in the signal and learn sparse representations. We also investigate the impact of incorporating fairness constraints into our architecture. » But since (as discussed in 2.1) there lacks evidence of the soundness of the fairness regularizer, combined with the fact that there is no evidence demonstrating that « regularization techniques encourage the concepts to be distinct, so they can span the entire input vector and be specialized » (as discussed in 1.4), or at least that the « different schemes for improving the interpretability of the concepts learned » truly are responsible for such observations, the soundness of all Contribution 2 can be questioned.```
>
> **Fairness Regularizer:**
>
> Based on the reviewer’s suggestion, we have provided additional results in the supplementary comparing model performance before and after the fairness regularizer is applied. Previously, we had only presented the FNR/FPR values for each subgroup to show the performance of the majority sensitive group doesn’t deteriorate as a result of the regularization. These results can be found in Section G of the supplementary. We also refer the reviewer to our previous answer discussing the efficacy of the fairness regularizer for more details.
>
> **TANGOS Regularizer:**
>
> The interpretability of the feature space in deep learning problems can be approached from various angles. The notion of interpretability we have focussed on is that the concepts are distinct and specialized and also span the entire input vector. TANGOS regularization assisted us in achieving this by quantifying the contribution of each input dimension to a particular concept. We examine the gradient of each concept obtained from the concept extractors with respect to the input signal. The TANGOS loss consists of two components: The first component enforces sparsity, emphasizing a concentrated subset of the input vector for each concept. The second component promotes uniqueness, minimizing the overlap between the input subsets from which each concept is derived. Sparsity is achieved by taking the L1-norm of the concept gradient attributions with respect to the input vector. To promote decorrelation of signals learned in each concept, the loss is augmented by incorporating the inner product of the gradient attributions for all pairs of concepts. Additional details about the mathematical formulation of TANGOS have been provided in Section F.1 of the appendix. The desired level of interpretability can be adjusted by varying the relative weights of these terms with respect to the probabilistic checklist objective. In Section F.2, we show how the model performance varies in terms of accuracy, precision and recall by tweaking these weights. The section elegantly captures the trade-off between interpretability and model performance. Furthermore, we plot the images and corresponding gradient attributions heat maps for seven input samples of the Image 2 modality of the MNIST dataset for different combinations of sparsity and correlation regularization terms. This plot can be found in Section F.2 (Figure 5). Similar analysis has also been done for MIMIC Clinical Time Series (Section F.3, Figure 8). It is evident from both these plots that the gradient attributions for both the concepts was identical when TANGOS regularization is not used (i.e \lambda_sparity = 0 and \lambda_correlation = 0). As the regularization weights are gradually increased, the gradient attributions start to diverge, yielding distinct concepts. We agree with the reviewer that this assessment relies on visual inspection because it’s hard to quantify interpretability. Nevertheless, a significant advantage of our approach lies in its flexibility, enabling users to experiment with other notions of interpretability tailored to different applications.
>
> We also found it easier to interpret our results on NLP Medical Abstract classification without the need of TANGOS regularizer. We represented our learnt checklist in Figure 1. This is because the building blocks of text are tokens which are inherently human understandable, on the other hand it’s much harder to comprehend pixel-level RGB intensities.
>
> ```Typos: ```
>
> We thank the reviewer for spotting the typing mistakes. We have proofread the paper carefully and resolved the listed issues.

---

> > ### Comment · Reviewer_SxKf · 2023-11-20
> >
> > I thank the authors for their exhaustive response. I am globally satisfied with the answers to my concerns. However, I still think  gradient attributions is insufficient for interpreting the concepts, as said before, for it is one explainability technique that is far from being reliable. Therefore, I see this work relevant in areas where the inputs of the problems are intrinsically interpretable. And even though the scope, from this perspective, is more restrained, the probabilistic approach and the fairness constraints are in themselves valuable contributions; I will raise thus my score. Thank you.

---

### Official Review · Reviewer_Ucnh · 2023-10-31

**Soundness:** 2 fair
**Presentation:** 2 fair
**Contribution:** 2 fair
**Rating:** 3
**Confidence:** 4

**Summary:**

The paper proposes a novel method based on probabilistic logic programming to learn predictive checklists from diverse data modalities including images and time series. The proposed approach was validated using several public benchmark datasets.

**Strengths:**

- Originality: The paper demonstrates originality in creative combinations of existing ideas and approaches to the target problem
- Clarity: Problem formulations and related works are clearly described and cited. The paper is well-organized with most components including limitations.
- Significance: Classification performances are reported with multiple metrics and confidence intervals

**Weaknesses:**

1. One weakness is the results discussion using MNIST data only, which is not so intuitive in the checklist concept motivated by healthcare examples in the introduction part. And the paper has results from clinical data of PhysioNet and MIMIC III in the supplementary materials, which should be much better than the MNIST story. The necessity of using checklist, instead of other benchmark methods, on the experiment data tasks (especially non-healthcare MNIST data) is another question not explained.
2. Model comparison in Table 1 would be better to also be illustrated in graph and plots for easy visualization.
3. Concepts learned from images seem not human-interpretable if looking at the example in Figure 3. The two concepts might still look like visual patterns that could be only differentiable by machines or algorithms. It will be hard in practice to create human-understandable checklist out of the concepts illustrated, especially in clinical domain.
4. Concepts learned from other data modality is not illustrated in the main paper, especially time series and text, which weakens the claim of interpretation utility of the proposed algorithm in different modality.
5. Several typos in the paper, e.g. (683, 2021) on page 6, not sure whether it's citation or time series specification; and also "We create a We briefly" in line #2 on page 7. The paper needs some proofreading.

**Questions:**

1. Same in weakness. If the checklist concept learned from the data is not easy for human to understand and annotate, what's the potential utility of the proposed method?
2. How does the proposed method compared to other benchmark method without using checklist? a.k.a. Why using checklist to identify MNIST or predict sepsis or mortality? Is the performance better than other methods in the literature?

---

> ### Author Response · Authors · 2023-11-20
> **Rebuttal by Authors**
>
> We thank the reviewer for their valuable comments and insights.
>
> ``` 1) Model comparison in Table 1 would be better to also be illustrated in graph and plots for easy visualization. ```
>
> We thank the reviewer for this suggestion. We have added this plot to Section I of the appendix to help the readers.
>
> ``` 2) Concepts learned from other data modality are not illustrated in the main paper, especially time series and text, which weakens the claim of interpretation utility of the proposed algorithm in different modality.```
>
> We direct the reviewer to Figure 1 of the paper, which illustrates the checklist learnt from the medical abstract classification task. As previously explained in the global response, understanding results for NLP tasks is significantly simpler due to the comprehensibility of tokens. We have now included a description of the technique used for generating this checklist in the main paper. While experiments involving the TANGOS regularizer on time series datasets have been conducted, we have presented them in the supplementary section. This decision was made as these results require assessment by domain experts to discern patterns effectively.  Instead, our focus has shifted to the synthetic MNIST dataset, as its known ground truth concepts make it easier to validate our approach. We also discuss the results of the PhysioNet Tabular dataset.
>
> ``` 3) Same in weakness. If the checklist concept learned from the data is not easy for human to understand and annotate, what's the potential utility of the proposed method? ```
>
> We direct the reviewer to our discussion on interpretability of concepts and utility of the method in the global response.
>
> ``` 4) How does the proposed method compared to other benchmark method without using checklist? a.k.a. Why using checklist to identify MNIST or predict sepsis or mortality? Is the performance better than other methods in the literature?```
>
> We have included the following baselines in Table 1:
> - **ML (non-checklist) baselines:** LSTM/CNN/BERT + MLP (for 10 selected features and all features in the dataset), LSTM/CNN/BERT + LR.
> - **Checklist baselines:** Unit weighting, ILP mean thresholds, MIP Checklist.
>
> The CNN/LSTM + MLP for MIMIC which is trained on all the features in the dataset acts as an upper baseline to quantify the performance loss incurred by switching to an interpretable checklist classifier.
>
> Through these experiments, we aim to showcase that ProbChecklist surpasses existing checklist methods and achieves comparable performance to MLP (non-interpretable) methods. It’s important to note that a checklist, due to its binary weights, has a strictly lower capacity and is less expressive than deep learning but possesses a more practical and interpretable structure. Despite this, it exhibits similar performance to an MLP.
>
> ```5) One weakness is the results discussion using MNIST data only, which is not so intuitive in the checklist concept motivated by healthcare examples in the introduction part. And the paper has results from clinical data of PhysioNet and MIMIC III in the supplementary materials, which should be much better than the MNIST story. The necessity of using checklist, instead of other benchmark methods, on the experiment data tasks (especially non-healthcare MNIST data) is another question not explained.```
>
> We direct the reviewer to the global response 1 and answers to questions 1 and 4 above.
>
> ``` 6) Several typos in the paper, e.g. (683, 2021) on page 6, not sure whether it's citation or time series specification; and also "We create a We briefly" in line #2 on page 7. ```
>
> The paper needs some proofreading. We thank the reviewer for spotting the typing mistakes. We have proofread the paper carefully and resolved these issues.

---

> ### Author Response · Authors · 2023-11-22
> **We are available to answer any further comments.**
>
> Dear Reviewer,
>
> Thank you again for reviewing our paper and for your encouraging feedback!
>
> Have all the concerns you raised been adequately addressed? We are glad to provide you with complementary responses.
>
> Thank you very much.
>
> Best Regards,
> The authors

---

### Official Review · Reviewer_eMqU · 2023-11-01

**Soundness:** 2 fair
**Presentation:** 2 fair
**Contribution:** 2 fair
**Rating:** 6
**Confidence:** 4

**Summary:**

This paper proposes a method for learning checklist models for diverse data modalities, such as images, time series, and text. Checklists are a type of interpretable models that are widely used in clinical settings. A checklist model consists of a set of concepts, each of which is assigned an integer weight (always +1 in this paper). The prediction is made by summing the weights of the concepts that are present in the input and comparing the sum to a threshold $T$. Existing methods for learning checklist models are limited to tabular data. To learn checklists from these raw data modalities, the authors propose to train neural networks using a probabilistic logic programming (PLP) framework. Basically, the neural network maps the input signals to a fixed number of logits, each of which is regarded as the log probability of the presence of a concept. One can then use the logits to compute the likelihood of the positive/negative label based on the definition of the checklist model. The likelihood of the positive label is the probability of the event that at least $T$ concepts are present in the input. The model is then trained with the cross-entropy loss. The authors also propose to add several regularization terms to encourage interpretability and fairness. In the experiments, the proposed model is compared to integer programming and deep learning baselines.

**Strengths:**

- Originality: This work extends checklist learning to data modalities other than tabular data and combines the power of deep learning with the interpretability of checklist models. The proposed method is interesting.
- Clarity & Quality: The background and methodology are clearly presented. The paper is easy to follow.
- Significance: The proposed method seems to be a practical solution to the problem of learning checklist models from raw data modalities. Such models, if learned successfully, may be used in many real-world applications, such as clinical decision support.

**Weaknesses:**

- There are too many hyperparameters in the proposed method, including the weight of the regularization terms, the number of concepts, and the threshold $T$, in addition to the architecture details of the neural networks. The authors should provide some guidance on how to choose these hyperparameters.
- The learned "concept"s are hard to interpret from my point of view. The authors suggest that the concepts can be sensed by using post hoc attribution methods. However, it is well-known that the attribution methods are not perfect and may not be reliable.
- Missing related work: I believe this work should be connected to the literature on concept-based explanation and learning, such as [1], [2], and the references therein. The authors should discuss the connections and differences.
- The computational cost of the proposed loss function scales exponentially with the number of concepts.
- Typos: "LSTMS" -> "LSTMs", "TANGOS" -> "TANGOs"

[1] Amirata Ghorbani, et al. Towards Automatic Concept-based Explanations. NeurIPS 2019
[2] Pang Wei Koh, et al. Concept Bottleneck Models. ICML 2020

**Questions:**

- Is it possible to extend the proposed method to learn checklist models with integer weights that are not necessarily +1? This may be useful in many applications.

---

> ### Author Response · Authors · 2023-11-20
> **Rebuttal by Authors (1/2)**
>
> We thank the reviewer for their valuable comments and insights.
>
> ```1) There are too many hyperparameters in the proposed method, including the weight of the regularization terms, the number of concepts, and the threshold, in addition to the architecture details of the neural networks. The authors should provide some guidance on how to choose these hyperparameters. ```
>
> We thank the reviewer for this suggestion. We had included a section on hyperparameter tuning in the appendix (Section C). We have now added a reference to it in the main paper for the interested readers.
>
> The ProbChecklist framework allows experts to design user-centric checklists. Hyperparameters $d_k, \ T and \ M$ represent the structure/compactness of the checklist but alone aren’t sufficient to garner information about the checklist’s performance. Different $d_k$ are tried for each modality in an increasing fashion to find the one that performs best. Sensitivity analysis to study the relation between $d_k$ and performance (Figure 3a) suggests that the performance saturates after a certain point. This point can be determined experimentally for different modalities. M, total concepts in the checklist, is obtained by pruning concepts that are true for insignificant samples. Our experiments showed that pruning was not required since all the concepts were true for a significant fraction of samples. This indicates the superior quality of the concepts.
>
> We try different values of T in the range [M/4, M/2] (total items M) to find the most performant model. However, this hyperparameter tuning doesn’t contribute to the computational cost. For $d_k$, we only only 2-3 values, and use the same value for all the features. Domain experts will be more equipped to choose these values based on their knowledge of the features. For example, if we are recording a time series feature known to stay stable (not fluctuate much), then low $d_k$ is sufficient. The value of $d_k$ also depends on the number of observations (most blood tests aren’t performed hourly, but heart rate and oxygen saturation are monitored continuously).
>
> ```2) The learned "concept"s are hard to interpret from my point of view. The authors suggest that the concepts can be sensed by using post hoc attribution methods. However, it is well-known that the attribution methods are not perfect and may not be reliable. ```
>
> We direct the reviewer to our discussion on interpretability of concepts in the global response.
>
> ``` 3) Missing related work: I believe this work should be connected to the literature on concept-based explanation and learning, such as [1], [2], and the references therein. The authors should discuss the connections and differences. [1] Amirata Ghorbani, et al. Towards Automatic Concept-based Explanations. NeurIPS 2019 [2] Pang Wei Koh, et al. Concept Bottleneck Models. ICML 2020  ```
>
> We thank the reviewer for recommending these papers. Concept Bottleneck Model (CBM) is an approach to make deep learning architectures more interpretable by adding a concept layer before the last fully connected layer. Each neuron in this layer represents a human understandable concept. One major limitation of this technique is that annotated data for predefined concepts is required which is expensive to collect. We have included these papers in our related works section.
>
> ``` 4) The computational cost of the proposed loss function scales exponentially with the number of concepts.```
>
> We offer a comparison of training times with the MIP checklist method on the PhysioNet Tabular dataset to evaluate the extent of this limitation (Section B of the appendix). It's crucial to highlight that while MIP Checklist performs effectively with tabular data, successfully uncovering the optimal solution, its performance is poor when applied to MNIST synthetic setup. Even when we set the runtime for Gurobi solver as 1 hour, it struggles to achieve optimal solutions for many cases. On the other hand, ProbChecklist stands out as more reliable, capable of performing end-to-end training and successfully learning the optimal solution.

---

> ### Author Response · Authors · 2023-11-20
> **Rebuttal by Authors (2/2)**
>
> ``` 5) Is it possible to extend the proposed method to learn checklist models with integer weights that are not necessarily +1? This may be useful in many applications. ```
>
> We warmly thank the reviewer for this very valuable suggestion. Before we formulate a method to achieve this, it is important to note that this would make it harder to interpret the checklist. More specifically, the meaning of the checklist threshold used for classification of samples would now change. Previously, it represented the minimum number of true concepts for a positive classification. Now it signifies a score - the minimum value of the weighted sum of concept probabilities for a positive classification.
>
> We propose the following extension to allow for integer weights larger than 1. Given K data modalities as the input for sample i, we train K concept learners to obtain the vector of probabilistic concepts  of each modality $\mathbf{p_i^k} \in [0,1]^{d'}$. Next, we concatenate the full concepts probabilities ($\mathbf{p_i}$) for sample i. At this point, we can introduce a trainable weight vector $W \in [0,1]^{d'}$ (with $\sum_{i=1}^{d’}w_i = 1$) of the same dimension as $p_i$ (concept probabilities) which will capture the relative importance of the features. Element-wise product ($\circledcirc$) of $p_i$ and W represents the weighted concept probabilities and can be denoted with $Wp_i$. This vector can be normalized by dividing each element with the maximum entry (L0 norm). While training it i For training the concept learners, we pass $Wp_i$ through the probabilistic logic module. After training, the integer weights corresponding to each concept in the checklist can be obtained by converting the W to a percentage: $W_{int}$. At inference time, we discretize $\mathbf{C_i}$ to construct a complete predictive checklist. Next, compute the score $W_{int}^T C_i$ and compare it against the checklist threshold, $M$, to classify the sample.
>
> We appreciate the reviewer's suggestion and concur that this extension of our method could be valuable for users. As a result, we have incorporated a figure and a concise description of the approach in the appendix (Section H).
>
>
> ``` 6) Typos```
>
> We thank the reviewer for spotting the typing mistakes. We have proofread the paper carefully and resolved them.

---

> > ### Comment · Reviewer_eMqU · 2023-11-22
> >
> > Thanks for your detailed response. I will update my rating to borderline accept. Please recheck the notation consistency in the revised submission. For example, I think the $d_k$ in Figure 3a should be $d'_k$ in Section 4.3. Proofreading is needed.

---

> > > ### Author Response · Authors · 2023-11-23
> > >
> > > We sincerely thank the reviewer for raising the score and for the encouraging feedback. We have rectified the notation error in Figure 3a to maintain consistency with Section 4.3. The paper has been updated after proofreading.

---

### Author Response · Authors · 2023-11-20
**Global Response (1/2)**

We want to thank all reviewers for their insightful comments that contributed to improving our paper. We are glad to report that we have implemented all the suggested changes in the paper.

We address here main topics that were raised in the reviews.

**1) Interpretability of checklist structure (checklist classifier)**

Reviewers eMqU, Ucnh, and SxKf have questioned the interpretability of the checklist structure. We want to further motivate the interpretability of a checklist, and of our classifier.
Although ProbChecklist uses a probabilistic objective for training the concept learners, the end classifier used for inference is indeed a discrete checklist. At inference time, the end classifier is thus a discrete checklist.
While this makes the classifier highly interpretable, it also shifts the focus of interpretability to the learnt concepts. We fully realize and acknowledge this trade-off. We do not claim to definitely solve the problem of interpretability but rather investigate the feasibility of an alternative approach. We recognize the problem of the potential non-interpretability of the concepts in our architecture. However, this is a structural issue from the field of interpretable machine learning in general. Nevertheless, we spend a significant effort in the experiment assessing the interpretability of the concepts, relying on several state of the art methods, as described below.

**2) Interpretability of learnt concepts**

Identifying patterns from the binarized concepts is largely based on visual inspection. To aid our analysis, we use gradient attribution of the concepts with respect to the input to identify parts that contribute to each concept.
We discuss the interpretability of the concepts for each modality separately:

- **Continuous Tabular Dataset (PhysioNet Sepsis Tabular):** Our method works effectively on continuous tabular data where we know what each attribute represents. ProbChecklist learns the thresholds to binarize these continuous features to give the concepts. These concepts are inherently interpretable. All existing methods are designed to operate on continuous or categorical tabular datasets only. As such, our method is already novel. Nevertheless, we investigated the ability of our architecture to handle more complex data modalities.

- **Image and Time Series Tasks (MNIST/MIMIC):** Our initial results highlighted that the gradient attributions for the different concepts learned from one modality were very similar (plots in Section F.2 of the supplementary).  Pixel intensities alone are insufficient for automatically interpreting the concepts, giving rise to the need for visual inspection by domain experts. Therefore, we opted to visualize the gradient attributions of the concepts concerning the input. These plots aid domain experts in extracting patterns and recognizing the learned concepts. TANGOS enforces sparsity and decorrelation among concepts, thereby specializing them to specific input regions and preventing redundancy. While this represents one notion of interpretability, different applications may benefit from alternate definitions suited to their needs. One significant benefit of our approach is its adaptability to incorporate various other interpretability methods, enhancing its flexibility.

- **NLP Tasks (Medical Abstract Classification):** Compared to images and time series, interpreting concepts learned from textual data is easier because its building blocks are tokens which are already human understandable. In this setup, instead of employing TANGOS regularization, we identified words associated with positive and negative concepts (positive and negative tokens). Each concept is defined by the presence of positive words and the absence of negative words. The resulting checklist is visualized in Figure 1. We have edited the main paper to include more details about this.

---

> ### Author Response · Authors · 2023-11-20
> **Global Response (2/2)**
>
> **3) Motivation and Utility of the proposed method**
>
> - All existing approaches for checklist learning are tailored for tabular datasets where the concepts are predefined. Our method not only operates effectively on continuous tabular datasets but also provides the flexibility to extend to complex modalities such as text, images, and time-series, thereby establishing its superiority. To ensure a fair comparison with previous methods, we conduct experiments using the PhysioNet Sepsis Tabular dataset. The results demonstrate that ProbChecklist achieves performance comparable to existing methods. Notably, ProbChecklist learns thresholds for binarizing continuous features to derive concepts, distinguishing it from all other techniques except MIP Checklist, which need binarized features.
> - As mentioned above, for our experiments on time series and image datasets we focus on one notion of interpretability: that the learnt concepts should be distinct and should span the entire input vector. These constraints were enforced by employing the TANGOS regularizer. However, different applications may necessitate alternative definitions of interpretability, which can be seamlessly integrated into ProbChecklist's framework.
> - The bulk of data collected today comprises images, text, and time series information. It's imperative to shift our attention toward devising methods to handle non-tabular data. We realize the shortcomings of our approach, but this is the first step towards learning checklists from complex modalities.
>
> **4) Additions to the paper**
>
> In response to the valuable feedback provided by the reviewers,  we have included additional results.
> - We add an ablation study in the supplementary (Section B) comparing the training times of ProbChecklist and MIP Checklists on the PhysioNet Tabular Dataset. We also elaborate on situations where ProbChecklist was more effective than MIP checklists at learning checklists in the MNIST setup.
> - We expand the fairness results in Section G of the supplementary to include the model performance before and after fairness regularizer is applied.
> - We include a discussion on the interpretability of concepts for the Medical Abstract Dataset in Section 5.2 of the main paper.
> - In our current method, each item in the checklist holds a weight of +1. We present an approach to expand ProbChecklist, enabling the learning of integer weights for each item. This extension has the potential to broaden the applicability of our method. He describe the approach in Appendix H.

---

### Meta-Review · Area_Chair_5NzQ · 2023-12-08

**Metareview:**

The paper proposes to learn interpretable checklists from various modalities using probabilistic logic programming. This is an important and well-motivated problem, specifically in clinical context. The paper is well motivated, well-written, and the methodology seems sound.

Some reviewers remarked that the interpretability of the approach hinges on interpretability tools for deep learning tools, which need to be considered with care. The experiments are relatively limited.

The AC has as an additional comment: it is not clear where the logics part enter the method. In section 4.4, it is mentioned "The checklist prediction formula of Equation 3 an be understood as logical rules in a probabilistic logical program." But equation 3 appears in FAIRNESS REGULARIZATION and seems not to define any logical part of the problem. The link points to page 4, where the prediction is expressed as simple threshold. Is this the logic part of the method? Given the title, one would actually expect that some significant logical knowledge would enter the method.

**Justification For Why Not Higher Score:**

While the paper is well-motivated, there substantial criticisms remain. The paper is probably rather borderline, but probably too weak for ICLR.

**Justification For Why Not Lower Score:**

NA

---

### Decision · Program_Chairs · 2024-01-16

Reject